# Atmospheric pollution over the eastern Mediterranean during summer – A review

Uri Dayan[1], Philippe Ricaud[2], Régina Zbinden[2] and François Dulac[3]

[1]Department of Geography, The Hebrew University of Jerusalem, Jerusalem, 91905, Israel

[2]CNRM, Météo-France, CNRS UMR3589, Toulouse, France

[3]Laboratoire des Sciences du Climat et de l'Environnement (IPSL-LSCE), CEA-CNRS-UVSQ, Univ. Paris-Saclay, Gif-sur-Yvette, France

*Correspondence to:* Uri Dayan (msudayan@mscc.huji.ac.il)

**Abstract.** The eastern Mediterranean (EM) is one of the regions in the world where elevated concentrations of primary and secondary gaseous air pollutants have been reported frequently, mainly in summer. This review discusses published studies of the atmospheric dispersion and transport conditions characterizing this region during the summer, followed by a description of some essential studies dealing with the corresponding concentrations of air pollutants such as ozone, carbon monoxide, total reactive nitrogen, methane and sulfate aerosols observed there.

The interlaced relationship between the downward motion of the subsiding air aloft induced by global circulation systems affecting the EM and the depth of the Persian Trough, a low-pressure trough that extends from the Asian monsoon at the surface controlling the spatio-temporal distribution of the mixed boundary layer during summer is discussed. The strength of the wind flow within the mixed layer and its depth affect much the amount of pollutants transported and determine the potential of the atmosphere to disperse contaminants off their origins in the EM. The reduced mixed layer and the accompanying weak westerlies, characterizing the summer in this region, lead to reduced ventilation rates, preventing an effective dilution of the contaminants. Several studies pointing at specific local (e.g. ventilation rates) and regional peculiarities (long-range transport) enhancing the building up of air pollutant concentrations are presented.

Tropospheric ozone ($O_3$) concentrations observed in the summer over the EM are among the
highest over the northern hemisphere. The three essential processes controlling its formation
(i.e., long-range transport of polluted air masses, dynamic subsidence at mid-tropospheric levels,
and stratosphere-to-troposphere exchange) are reviewed. Airborne campaigns and satellite-borne
initiatives have indicated that the concentration values of reactive nitrogen identified as
precursors in the formation of $O_3$ over the EM were found to be 2 to 10 times higher than in the
hemispheric background troposphere. Several factors favor sulfate particulate abundance over
the EM. Models, aircraft measurements, and satellite derived data, have clearly shown that
sulfate has a maximum during spring and summer over the EM. The carbon monoxide (CO)
seasonal cycle, as obtained from global background monitoring sites in the EM is mostly
controlled by the tropospheric concentration of the hydroxyl radical (OH), and therefore
demonstrates high concentrations over winter months and the lowest during summer when
photochemistry is active. Modeling studies have shown that the diurnal variations in CO
concentration during the summer result from long-range CO transport from European
anthropogenic sources, contributing 60 to 80% of the boundary-layer CO over the EM. The
values retrieved from satellite data enable us to derive the spatial distribution of methane ($CH_4$),
identifying August as the month with the highest levels over the EM. The outcomes of a recent
extensive examination of the distribution of methane over the tropospheric Mediterranean Basin,
as part of the Chemical and Aerosol Mediterranean Experiment (ChArMEx) program, using
model simulations and satellite measurements is coherent with other previous studies. Moreover,
this methane study provides some insights on the role of the Asian monsoon anticyclone in
controlling the variability of $CH_4$ pollutant within mid-to-upper tropospheric levels above the
EM in summer.

## 1 Introduction

The relationship between atmospheric air pollutant concentrations and large-scale atmospheric
circulation systems have been examined over the past decades (e.g., Davis and Kalkstein, 1990;
Dayan et al., 2008). This strong relationship and its issuing dispersion condition at several scales,
and climatically related variables such as air pollutants, is presented in this work as part of the
Chemistry-Aerosol Mediterranean Experiment (ChArMEx; http://charmex.lsce.ipsl.fr).
However, a first major drawback in attributing air pollutant concentrations to variations in large-
scale atmospheric circulation arises from the fact that changes in removal processes and upwind
emissions are not necessarily concurrent with variations in circulation. Some efforts were
undertaken, mainly through coupled chemistry-climate models to treat and analyze at the same
time, the changes in general circulation and atmospheric chemistry (Hein et al., 2001; Dastoor
and Larocque, 2004). Moreover, secondary pollutants such as tropospheric ozone ($O_3$) result
basically from photochemical reactions among precursors and, as such, are controlled by air
mass characteristics such as temperature and humidity, and cloud cover/solar radiation.
Accordingly, changes in trace gases concentrations are modified with respect to exposure of the
differing air masses driven by changes in atmospheric circulation.
A second substantial shortcoming in trying to associate changes in pollutant concentration to
variation in circulation patterns is their different life span and distribution. For example, durable
greenhouse gases (GHG) such as methane ($CH_4$) and carbon dioxide ($CO_2$) are characterized by
long life times of years as compared to nitrogen oxides and aerosols which are most relevant for
short spatial and temporal scales (Andreae, 2001; Voulgarakis et al., 2010). Radiative forcing of
aerosols is of much higher spatial variability than GHG forcings due to the relatively short
aerosol lifetime (daily-weekly scale) compared to that of GHG (monthly-yearly scale).
Both natural and man-made factors converge over the EM favoring the accumulation of pollutant
concentration during summer. This region is in the crossroad of both large-scale convective
motions: Hadley and Walker cells leading to subsidence. This process results in a reduced
mixing depth, which inhibits an efficient dispersion of the pollutants. Moreover, the EM is a
hotspot of high solar radiation driving the photochemistry of the atmosphere. In addition, the
summer prevailing westerlies at shallow tropospheric layers favor the transport of pollutant-
enrichedair masses from central and eastern Europe to the eastern Mediterranean (EM). Based on
the above key factors, this review focuses explicitly on summertime. Lelieveld et al. (2002)
studied air pollutant transport over the EM in summertime. They report that the synoptic flow is
controlled by the strong east-west pressure difference between the Azores high and the Asian
monsoon low, with additional influence in the upper troposphere from the Tibetan anticyclone.
This yields a contrasted situation in the tropospheric column with European influence in the
lowermost troposphere, a much longer-range transport from Asia and North America at mid-
tropospheric levels, and a major impact from Asia in the upper troposphere and lower
stratosphere.
Desert dust is abundant over the EM, transported from two major source regions: the North
African Sahara and the Arabian deserts. However, in general and predominantly, mineral dust
affects the EM during all seasons except summer (e.g., Dayan et al., 1991; Moulin et al., 1998;
Sciare et al., 2003), reason why mineral dust is not in the scope of this study which is focused on
summer conditions.
In this review, we first describe the atmospheric dynamic conditions favoring the building up of
tropospheric air pollutant concentrations. Secondly, we propose a synthesis of the essential
studies on air pollutant concentrations including $O_3$, sulfate ($SO_4$) aerosols, total reactive
nitrogen ($NO_y$), carbon monoxide (CO), and $CH_4$. The sources of the data reported include in-
situ observations, balloon-sounding, aircraft and space-borne observations as well as model data,
which results, in terms of dynamics, are mostly updated over 1948-2016 on availability.

## 2 Summer atmospheric dynamic conditions favoring the building up of tropospheric pollutants concentrations

Different spatial and temporal scales of motion affect pollutant transport and dispersion: the
microscale, mesoscale, synoptic scale, and macro-, or global scale. At the scale of a few months,
the planetary boundary layer is relatively well mixed. However, on shorter timescales and near
the Earth's surface (where pollutants are emitted), transport and dispersion are often limited by
atmospheric conditions. In this section, we will focus on the global and synoptic scale processes
that favor a potential accumulation of pollutants in the EM troposphere.

### 2.1 Global and synoptic scales inducing subsiding conditions over the eastern Mediterranean

In general, the atmospheric conditions over the EM are persistent during the summer and subject
to two essential processes. The first is the cool advection at shallow tropospheric layers caused
by the strong, dry north Etesian winds generated by the east–west pressure gradient manifested
by large scale circulation features, low pressures over the EM as an extension of the Persian
Trough (PT) and the high pressure over central and southeastern Europe (Tyrlis and Lelieveld,
2013). This surface low pressure trough extends from the Asian monsoon through the Persian
Gulf and further, along southern Turkey to the Aegean Sea (Figs. 1 and 2). The second is the
dynamic subsidence generated by several global-scale processes: the African Monsoon as part of
the subtropical descending branch of the Hadley cell (Fig. 3 left), the Asian Monsoon as part of
the Walker cell (Fig. 3 right) and subsidence caused by the negative relative vorticity
characterizing this region, during summer, as explained further on.
Rodwell and Hoskins (1996) used a hydrostatic primitive equation model initialized by a six-
year June to August climatology derived from the European Center for Medium Range Weather
Forecasts (ECMWF) analyses to investigate the monsoon desert mechanism enhancing
summertime descent in the Mediterranean subtropics. They argued that the subsidence center in
the EM is governed by the Asian monsoon rather than by the Hadley circulation and explained it
by diabatic heating in the Asian monsoon region that induces a Rossby wave to its west, which
generates air masses descent. This adiabatic descent balances the horizontal advection on the
southern flank of the mid-latitude westerlies. Among the summertime descent regions, the
strongest is located over the EM. Initiation of the descent over the EM coincides with the
northward movement of heating during the onset of the monsoon. The anticyclonic center over
north-west Africa and the monsoon result in an adiabatic warming that reduces the specific
humidity and consequently enhances further the descent due to diabatic radiative cooling under
cloudless sky conditions. Moreover, trajectory calculations performed by Rodwell and Hoskins
(1996) revealed that the bulk of the sinking air masses originate from mid-latitude regions rather
than over the intense monsoon convection areas over northern India. This is consistent with
Tyrlis et al. (2013) who analyzed the thermodynamic state over the EM and calculated the
temperature changes caused by horizontal advection by using ECMWF forecasts for diabatic
heating over this region. They found that subsidence at mid and lower levels is primarily driven
by the midlatitude westerly flow. Furthermore, Tyrlis et al. (2013) pointed at the steep slopes of
the isentropes in the free troposphere caused by the westward migration of the mid and upper
level warming of the atmosphere away from the diabatic heating sources, which further enhances
subsidence over the EM.
However, subsidence is neither restricted to mid-tropospheric levels nor solely associated to the
descending branch of these both general circulation cells. In summer, at higher atmospheric
layers, air masses converge and subside over the EM as contributed by an anticyclonic curvature
caused by anticyclonic centers formed over the Balkans. Such centers cannot be considered as
extensions of the Azores high since they exhibit typical warm-core high structures from the
surface up to mid-tropospheric levels (Anagnostopoulou et al., 2014). Tyrlis and Lelieveld
(2013) point at wave disturbances originating over the North Atlantic that activate intense ridges
over the Balkans. These ridges are further amplified by anticyclonic vorticity advection from
northwestern Africa and in tandem with diabatic cooling under clear skies form such centers over
central and southeastern Europe. The second dynamic factor inducing subsidence is an
anticyclonic wind shear as related to the position of the Subtropical Jet. Under these
circumstances, the southeastern part of the EM is exposed to the southern flank of the jet and
therefore prone to negative shear vorticity. Although shear vorticity is an order of magnitude
smaller than planetary vorticity, nearby jet streak makes this relative vorticity component
significant due to the strong change in wind speeds across the jet. Contribution of both
components enhances negative vorticity resulting in a total long-term mean negative vorticity of
-1 to -3 $10^{-5}$ s$^{-1}$ at 200 hPa (~12 km above sea level; a.s.l.) featuring the summer season over the
EM (Fig. 4).
The contribution of the above-mentioned dynamic subsidence generated by all processes results
in positive Omega values, defined as the Lagrangian rate of change in pressure with time,
indicating a downward air motion over the whole EM with its highest core of maximum
subsidence over Crete as depicted over mid-tropospheric levels (500 hPa geopotential height)
(Fig. 5).
Following the subsidence caused by the large-scale downward motion, the warming and drying
up is manifested by the delimiting sharp decrease in relative humidity over the EM Basin (Fig.
171 6).

Based on National Centers for Environmental Prediction/National Center for Atmospheric
Research (NCEP/NCAR) reanalysis for 2000–2012, Lensky and Dayan (2015) have recently
shown that the coincidence of negative vorticity advection aloft accompanied by cold horizontal
advection, at lower tropospheric levels, featuring the EM during PT synoptic conditions drive the
wind flow out of the thermal wind balance inducing a vertical downward motion (Figs. 2 and 7).
Ziv et al. (2004) found that the cool advection associated to the PT (Fig. 2) and the subsidence
related to both descending branches of the African and Asian monsoons (Fig. 3) are interrelated
and tend to balance each other. They suggest that this compensation mechanism explains the
reduced day to day temperature variations over the EM in summer (Fig. 8).
However, this monotonic regime is interrupted by the occurrence of hot day events resulting
from an expansion of the Subtropical High from North Africa towards the EM, which are prone
for elevated concentration of air pollutants. Harpaz et al. (2014) found that such episodes are
confined to the lower 4 km and controlled by the intensity of the negative temperature advection
rather than by the prevailing subsidence.

## 2.2 Atmospheric dispersion conditions over the eastern Mediterranean

The vertical velocity involved in the mixing process within the turbulent layer near the surface
and specifically its depth are important parameters in determining air pollutant concentrations at
shallow tropospheric levels (Zhang and Rao, 1999). The changes in the mixing layer depth
(MLD, i.e. the height of the convective atmospheric boundary layer marked by the base of a
thermal inversion) is governed by several factors: surface heating (Holtslag and Van Ulden,
1983), horizontal advection determined by the intensity of the sea breeze in coastal areas
(McElroy and Smith, 1991; Lensky and Dayan, 2012), local terrain over the continent (Kalthoff
et al., 1998), and the strength of the subsiding atmospheric air mass capping the mixed layer,
defined by the temperature profile within this stable layer and synoptic scale vertical motion
(Dayan et al., 1988). Beside these factors, the MLD is controlled also by thermal advection
associated with synoptic weather systems and therefore, develops under strong forcing by
synoptic scale circulations (Businger and Charnock, 1983; Holt and Raman, 1990; Sinclair et al.,
2010). Consequently, both the surface synoptic systems and their associated upper tropospheric
conditions should be taken into consideration for understanding the behavior of the MLD over
the EM basin and its adjacent coastal region.
Within the EM, numerous studies on the relationship between synoptic circulation and the
structure of the MLD over the continental EM were conducted in Israel, the southeastern part of
the basin. In particular, several studies were undertaken to characterize the spatial and temporal
behavior of the MLD (Neumann, 1952; Halevy and Steinberger, 1974; Rindsberger, 1974, 1976;
Dayan et al., 1988; Glaser et al., 1993; Lieman and Alpert, 1993; Dayan et al., 1996; Dayan and
Rodnizki, 1999; Dayan et al., 2002; Ziv et al., 2004) using sounding measurements at the Israel
Meteorological Service permanent site in Beit-Dagan (31.99°N, 34.82°E, 39 m a.s.l.), 8 km
southeast of Tel Aviv and at other sporadic sounding sites.
The atmospheric noon-time mixed layer during the summer over the EM region is featured by a
persistent elevated inversion base formed by a clear boundary line separating two differing air
masses, a cool and humid mass above ground capped by a much warmer and subsiding dry air.
The MLD is controlled by the interlaced relationship between the downward motion of the
subsiding air aloft and the depth of the PT at the surface (Fig. 9).
Due to the existing correlation between the MLD featuring the PT and air pollution episodes
over the EM evidenced in previous studies (Dayan and Graber 1981; Dayan et al., 1988; Koch
and Dayan, 1992), this barometric system was classified into three essential types (Fig. 10)
defined by the surface-pressure difference between Nicosia (35.16°N, 33.36°E, 149 m a.s.l.) in
Cyprus and Cairo (30.1°N, 31.4°E, 75 m a.s.l.) in Egypt, and the temperature at 850-hPa in Beit-
Dagan (Israel): Moderate PT, Shallow PT, and Deep PT (for details see Dayan et al., 2002).
Analyses of upper air measurements carried out regularly at Beit-Dagan, in the central coastal
plain of Israel, point at significant differences of the MLD for the several modes of the PT. The
overall summer mean noon time mixing depth values for 1981-1984 is 764 ±320 m (Dayan et al.,
1988). A classification with respect to the modes defined above resulted in mean and standard
deviation values of 428 ±144 m and 1010 ±214 m for the shallow and deep PT modes
respectively (Koch and Dayan, 1992). The spatial distribution of the mixing depth is rather
homogeneous under deep PT conditions over the central coastal plain of Israel as compared to
the shallow mode where its value is kept almost uniform above sea-level while penetrating
inland. Due to the important implication of this behavior on the building up concentration of air
pollutants, the lateral variance of the mixing depth was tested for part of the upper air
measurements performed at 4 sites concurrently during the 1981-1984 campaign (Dayan et al.,
1988). These sites on a west-east transect were: Nizanim (31.7°N, 34.63°E, 10 m a.s.l.) on the
southern coastal shore of Israel; Beit-Dagan (31.99°N, 34.82°E, 39 m a.s.l.) on the coastal plain;
Ruchama (31.5°N, 34.7°E, 210 m a.s.l.) ~20 km inland in the northern Negev Desert; and
Jerusalem (31.77°N, 35.21°E, 786 m a.s.l.). The average thickness of the mixed layer when
moving from the coast inland is reduced by 350 m while reaching Jerusalem (Fig. 11). The
longitudinal variance of the MLD North-South vertical cross section on 21 summer noon-time
upper air measurements performed simultaneously at 3 sites ~60 km apart along the Israeli coast
revealed that the MLD decreases gradually from north to south (Dayan et al., 1988). This finding
is explained by the greater distance of the southern sites from the cyclonic core of the PT (which
persists in summer to the northeast of Israel) as well as the decreased distance from the
anticyclonic center of the North African subtropical high (which persists during all seasons to the
southwest of Israel). This lateral and longitudinal variance indicates that the most reduced
summer MLDs are expected over the southeastern coast of the EM.
Most of the boundary layer studies from other coastal regions in the EM were conducted over the
Greek Peninsula and the Aegean Sea. Kassomenos et al. (1995) analyzed the seasonal
distribution of the MLD over the greater Athens area as obtained from the upper-air station of the
Greek Meteorological Service at the Hellinicon airport for the period 1974-1990. They point at a
noticeable annual variability in the afternoon MLD with maximum values (~800–1100 m a.s.l.)
being observed by the end of July. They explain these high values observed during summer and
the elevated inversion formed by the higher incoming solar radiation characterizing this season,
which is efficiently converted into sensible heat flux, favoring the development of a deep mixing
layer and the horizontal transport of warm air masses. Nevertheless, few summer days with
stably stratified atmosphere and very low MLDs (~300 m a.s.l.) inducing high surface pollution
levels in Athens's basin (Greece) were identified as well. This is consistent with Svensson
(1996) who analyzed such a summer day over the Athens's basin by applying a three-
dimensional coupled mesoscale meteorological and photochemical model. Tombrou et al. (2015)
mapped the MLD as part of the Aegean – GAME (Aegean Pollution Gaseous and Aerosol
airborne Measurements) for two summer days under Etesian flow conditions over the Aegean
Sea. The thermal profiles they analyzed demonstrate a well inflated MLD of 700 to 1000 m a.s.l.
during noon-time over Crete as compared to the shallow marine boundary layer (~400-
500 m a.s.l.) observed over both the east and west Aegean marine regions.
Characterizing the structure of the MLD spatial variation offshore over the EM basin is of
importance for getting a better insight on the processes, which control the dispersion of
contaminants over the sea. Few investigators (Gamo et al. (1982) and Kuwagata et al. (1990) for
Japan; Stunder and Sethuraman (1985) for the United States; Gryning (1985) for Denmark) have
analyzed the spatial variations of the atmospheric mixing layer in coastal areas. Similar studies
as related to the EM Basin are quite limited and deal also mainly on the conditions not directly
located over the open sea but rather at sites distant from the coastline.
In a 2006-2011 study based on a remote sensing tool, the ECMWF model and radiosonde
observations launched at Thessaloniki's airport (Greece, 40.6°N, 22.9°E, 10 m a.s.l.) ~1 km from
the coastline, Leventidu et al. (2013) found the MLD seasonal cycles peak with a summer
maximum of 1400, 1800, and 2100 m a.s.l. in June, July and August, respectively.
Much earlier in the unique study of this type we are aware of, Dayan et al. (1996) have evaluated
the spatial and seasonal distribution of the MLD over the whole Mediterranean Basin. Based on
~65000 air measurements from 45 radiosonde stations within and surrounding the basin from
spring 1986 through winter 1988, the MLD was derived from the potential temperature gradient
measured within the boundary layer and the capping stable layer above it. As expected, the
summer values prove to be generally higher over land and minimum over the most eastern and
western limits of the Mediterranean Basin (Fig. 12). They concluded that the distance from the
coastline and topography are the main factors influencing the spatial distribution of the MLD.
The steep gradient in MLD values observed as moving onshore is consistent with the elevated
summer values in Thessaloniki (Greece) reported by Leventidu et al. (2013).
Moreover, Dayan et al. (1996) found that the most striking temporal effect on MLD distribution
over the basin is caused by synoptic weather systems and the intensity of the sea-breeze along
the coast. The diminishing of the MLD over the Mediterranean Basin as moving from its center
eastwards toward the EM coast they have observed is consistent with the unique series of
measurements of the temperature profiles performed during the summer of 1987 near Ashdod
Harbor (31.82°N, 34.65°E), some 40 km south of Tel-Aviv (Israel) at 2 to 22 km from shore
using a tethered balloon where prominent inversion bases of 350 to 600 m a.s.l. were observed
(Barkan and Feliks, 1993). Moreover, such limited MLD values over the sea were obtained in the
airborne Gradient in Longitude of Atmospheric constituents above the Mediterranean basin
(GLAM) campaign in August 2014 (Zbinden et al., 2016): the MLD over the sea measured in the
period 6-10 August 2014 was approx. 800 m a.s.l. over Crete diminishing to about 400–500 m
a.s.l. over Cyprus.
The diurnal behavior of the MLD is assessed in the Israeli coastal plain based on routine
radiosonde ascents that are, unfortunately, of coarse temporal resolution. The hourly maximum
MLD is between 23:00 UTC and 05:00 UTC for all seasons and decreases gradually toward its
minimal value at 18:00 UTC (Dayan and Rodnizki, 1999)
However, since this cycle is governed mainly by synoptic weather systems, and the strength of
the sea-breeze, this behavior would be more significant for the summer. During this season, the
variation of the mixed-layer height due to diurnal variations of solar radiation and local terrain
effects is not obstructed by large-scale variations caused by frequent transitions between
different synoptic configurations, as featured by other seasons. Consequently, MLD variation is
most evident during the summer, mainly controlled by the daily sea-breeze cycle and heat fluxes
that are most intensive then. The layer minimal depth along the coast, of 760 m. a.s.l., is usually
observed during late afternoon hours when heat fluxes dissipate rapidly and the wind speed of
the cool sea breeze reaches its minimal rate. This process results in a decrease of the marine
turbulent boundary layer depth (Dayan and Rodnizki, 1999). These MLDs are less developed as
compared to the mean MLDs of 850 m. a.s.l. observed over the Athens basin by Kassomenos et
al. (1995).
Assessing the atmospheric dispersion conditions is commonly derived from the ventilation rates
calculation. This term is the MLD multiplied by the mean wind speed in the mixed layer,
representing the potential of the atmosphere to dilute and transport contaminants away from a
source region. Matvev et al. (2002) have calculated over 1948-1999 the mean and standard
deviation of the mixing depth, wind speed and long-term range of ventilation rates at the Israel
Meteorological Service sounding site in Beit-Dagan (Israel) for the summer. A criterion usually
adopted is that if the ventilation coefficient is less than $6000 \text{ m}^2 \text{ s}^{-1}$ the site has limited ventilation
(Dobbins, 1979; Pielke and Stocker, 1991).
Their results (Table 1) clearly show that the monthly long-term mean ventilation rates of
~$4500 \text{ m}^2 \text{ s}^{-1}$ characterizing the EM coastal zone during summer are reduced and therefore
inhibit an efficient dispersion of pollutants as compared to the summer mean values of
~7000 $m^2 s^{-1}$ obtained by Kassomenos et al. (1995) over the greater Athens area.

**2.3 Air mass origins over the eastern Mediterranean**

The chemical composition of an air mass is inevitably related to its origin and pathway.
Consequently, these both terms are indispensable in explaining its composition (Fleming et al.,
2012). Studies of the long-range transport (LRT) of pollution by trajectory models help us
interpreting and better defining the movement and removal processes affecting atmospheric
concentrations. Although changes in wind direction are observed on a diurnal and seasonal basis
depending on the synoptic conditions affecting the region, the prevailing wind flows over the
EM are from the west towards the east. Therefore, air pollutants emitted from upwind sources to
the west of the EM will reach the EM and will be added to those emitted locally. Indeed,
numerous observational and modeling studies have confirmed that the EM is affected by the
long-range transport of air pollutants originating from Europe (e.g., Dayan, 1986; Luria et al.,
1996; Wanger et al., 2000; Erel et al., 2002, 2007 and 2013, Matvev et al., 2002; Rudich et al.,
2008; Drori et al., 2012).
To get an insight on the LRT over the EM, the Air Resources Laboratory's trajectory model
(GAMBIT- Gridded Atmospheric Multi-Level Backward Isobaric Trajectories; Harris, 1982)
was applied over 1978-1982 (Dayan, 1986). The duration of each trajectory was chosen as 5-
days backward in time enabling the tracing of air masses originating from Europe, the
Mediterranean Basin, North Africa and the Near East close to the EM central coast of Israel (Fig.
342 13).

The 850-hPa level (~1500 m a.s.l.) was chosen as the most representative of the transport layer.
This level is selected as the intermediary level between the surface wind regime and the regime
of upper winds relatively free from local surface effects. Trajectory direction was divided into
five distinctive geographical classes as shown in Figure 13. Respective occurrences and seasonal
distributions can be summarized as follows:
1) Long fetch of maritime air masses from northwest Europe crossing the Mediterranean Sea,
accounting for 36%, was the most frequent on average and fell evenly throughout the whole
year;

2) Northeast continental flow that originated in eastern Europe, accounting for 30%, was the most frequent during the summer season;

3) Southeast flow from the Arabian Peninsula, accounting for 5%, was infrequent, occurring mainly during the autumn;

4a) Southwest flow along the North African coast, accounting for 11%, was the most frequent during late winter and spring; and

4b) South-southwest flow from inland North Africa was accounting for 7%, with a late winter and spring maximum.

Therefore, 1) and 2) trajectory types are indeed predominant with a summer maximum occurrence (>66%) over the EM coastal zone.

The 5-years (1983-87) flow climatology study of back trajectories at Aliartos, Greece (38.22°N, 23.00°E) revealed that about 40% of the 850-hPa back trajectories arriving to this site during summer, originate from northwest and north sectors (Katsoulis, 1999), which is consistent with the flow patterns reported by Kubilay (1996) for Mersin, Turkey. Katsoulis (1999) suggested that these predominant flow directions point at northeastern Europe and northwestern Asia as potential source regions.

These studies show that the main flow direction to the EM observed during summer lies between west and north wind sectors. This implies that the most probable source areas reaching and affecting the northern and eastern parts of this basin are the industrialized countries of eastern and central Europe located upwind to this part of the basin.

**3. Summer atmospheric air pollutant concentrations**

The EM is one of the regions in the world where elevated concentrations of primary and secondary gaseous air pollutants have been reported frequently. This region is influenced not only by local atmospheric dispersion conditions but also by the ability of the atmosphere to inherit a significant proportion of pollutants from European sources.

After reviewing the atmospheric dispersion and transport conditions characterizing the EM
during the summer, a summary of the essential results published over the last decade dealing
with trace gases and anthropogenic sulfate aerosol concentrations over this region is presented.
These studies demonstrate how the above described global and synoptic scale processes control
the extent of trans-boundary transport of air pollutants and chemical composition and
concentrations over the EM.
**3.1 Processes controlling $O_3$ formation**
Most tropospheric $O_3$ formation occurs when nitrogen oxides (NOx), CO and volatile organic
compounds (VOCs) react in the atmosphere in the presence of sunlight. Due to cloud free
conditions, high incoming solar radiation characterizes the EM during summer (Lelieveld et al.,
2002), which enhances the building-up of $O_3$ concentrations.
Numerous researchers have identified the EM as a "hot spot" of summertime tropospheric ozone
(e.g., Stohl et al., 2001; Roelofs et al., 2003; Zbinden et al., 2013; Zanis et al., 2014; Doche et al.,
2014; Safieddine et al., 2014).
Zbinden et al. (2013) derived the climatological profiles and column contents of tropospheric $O_3$
from the Measurements of Ozone by Airbus Aircraft program (MOZAIC) over the mid-northern
latitudes (24°N to 50°N) over the 1994-2009 period. Among the 11 most visited sites by the
MOZAIC aircrafts, is the EM cluster, which comprises 702 profile data from the two airports of
Cairo (31.39°E, 30.10°N, in Egypt), and Tel-Aviv, (34.89°E, 32.00°N, in Israel), from which
monthly means were derived. The $O_3$ volume mixing ratio obtained were converted to Dobson
units (DU) and validated against coincident ozonesonde profiles. Considering all sites, the EM
reaches the largest tropospheric $O_3$ column concentration of 43.2 DU in July that is related to an
extreme summer maximum within 1-5km, in agreement with the results derived from the space-
borne Ozone Monitoring Instrument (OMI) and Microwave Limb Sounder (MLS) by Ziemke et
al. (2011), pointing at the favorable photochemical conditions characterizing this region.
Zanis et al. (2014) identified a summertime pool with high $O_3$ concentrations in the mid-
troposphere over the EM over the 1998-2009 period as derived from the ERA-Interim reanalysis
$O_3$ data, the Tropospheric Emission Spectrometer (TES) satellite $O_3$ data, and simulations with
the EMAC (ECHAM5–MESSy) atmospheric chemistry–climate model. They indicated that the
high $O_3$ pool over the mid-troposphere is controlled by the downward transport from the upper-
troposphere and lower-stratosphere over this part of the MB, which is characterized by large-
scale subsidence. This subsidence is regulated by the Asian Monsoon as described in Section 2.1.
Furthermore, Zanis et al. (2014) based on previous case studies (e.g., Galani et al., 2003;
Akritidis et al., 2010) and climatological studies (e.g., Sprenger and Wernli, 2003; James et al.,
2003) and their own results deduced that the mechanism leading to high tropospheric $O_3$ over the
EM consists of two essential consecutive phases. In a first stage, an enrichment in stratospheric
$O_3$ occurs into the upper-troposphere via a stratosphere-to-troposphere transport process. In the
second stage, these $O_3$-rich air masses are transported downward by the strong summertime
subsidence characterizing this region.
Doche et al. (2014) analyzed tropospheric $O_3$ concentrations for the 2007-2012 period as
observed over the MB by the space-borne Infrared Atmospheric Sounding Interferometer (IASI).
They identified an abrupt west–east $O_3$ gradient in the lower troposphere over the Mediterranean
Basin with the highest concentrations observed over its eastern part. These concentrations were
observed at mid-tropospheric layers (3 km) caused by subsiding $O_3$-rich air masses from the
upper-troposphere, typifying summer. A clear and consistent seasonal variability emerges from
their measurements, showing a maximum of the 3-km partial column $O_3$ concentration in July
(Fig. 14). This is consistent with the study of Tyrlis and Lelieveld (2013) who found that the key
dynamic driving factors yielding to high $O_3$ concentrations in late July and early August, in mid
and lower free troposphere, are maximum tropopause folding activity, i.e., stratospheric air
intruding into the troposphere and the subsidence over the EM, featuring Etesian outbreaks,
which are temporally well correlated with the Indian monsoon. This tropopause folding is
manifested by a slightly lower tropopause in mid and lower free troposphere observed during
such outbreaks over the latitude of the Aegean forming a narrow "transport corridor" of positive
Potential Vorticity anomalies. Tyrlis and Lelieveld (2013) argue that such frequent subsidence of
high Potential Vorticity illustrates the important role of stratospheric intrusions in the summer
dynamic conditions over the EM. Furthermore, a climatology of tropopause folds over this
region based on the ERA-Interim data spanning the period 1979–2012 identified the Anatolian
plateau as hot spot of fold development that occurs ~25% of the time during July and August,
and a seasonal evolution linked with the South Asian monsoon (Tyrlis et al., 2014). The
contribution of tropopause folds in the summertime pool of tropospheric $O_3$ over the EM was
confirmed by Akritidis et al (2016) as simulated with the EMAC atmospheric chemistry model.
Based on IASI measurements and the Weather Research and Forecasting Model with Chemistry
(WRF-Chem), Safieddine et al. (2014) have shown that the air column of the first 2 km above
ground is enriched by anthropogenic $O_3$. Above 4 km, $O_3$ is mostly originating from outside the
Mediterranean Basin by LRT process or generated through stratosphere-to-troposphere exchange
characterizing the EM during the summer.
Air masses from surrounding regions in the EM atmosphere have a great impact on surface $O_3$
concentrations. In a recent study, Myriokefalitakis et al. (2016) have investigated the
contribution of LRT on $O_3$ and CO budget in the EM basin, using a global chemistry transport
model (CTM), the TM4-ECPL, driven by ECMWF Interim re-analysis project (ERA-Interim)
meteorology. They found that about 8% of surface $O_3$ concentrations are affected by local
anthropogenic emissions, whereas subsiding air masses from the free-troposphere and horizontal
transport from surrounding regions provide about 38% and 51% of $O_3$ sources, respectively, into
the EM mixed layer depth. Although elevated $O_3$ concentrations over the EM during the summer
are mainly attributed to LRT of polluted air masses originating from Europe and lingering over
the Mediterranean Basin, its enhancement as a secondary pollutant is also caused by its
precursors emitted along the coasts of the EM. Consequently, several studies dealing with $O_3$
concentrations measured over coastal sites surrounding the EM and its inland penetration are
presented.
Measurements of $O_3$ were performed at several sites in Crete and Greece and for rather long
periods: over the northern coast of Crete, at Finokalia (35.50°N, 26.10°E) 70 km northeast of
Heraklion, from September 1997 to September 1999 (Kouvarakis et al. 2000);  from a rural area
(40.53°N, 23.83°E) close to Thessaloniki in the north of Greece from March 2000 to January
2001 and from an $O_3$ analyzer installed in a vessel traveling routinely from Heraklion, Crete to
Thessaloniki, Greece, from Aug. to Nov. 2000. Based on these measurements, Kouvarakis et al.
(2002) pointed out the existence of a well-defined seasonal cycle in boundary layer $O_3$, with a
summer maximum both above the Aegean Sea and at Finokalia. They indicated that LRT is the
main factor accounting for the elevated $O_3$ levels above the EM. This finding is consistent with
the 1997–2004 surface $O_3$ time series at Finokalia (Crete) of Gerasopoulos et al. (2005) who
investigated the mechanisms that control $O_3$ levels and its variability. They identified transport
from the European continent as the main mechanism controlling the $O_3$ levels in the EM,
especially during summer when $O_3$ reaches a July maximum of 58 ±10 ppbv. Moreover, on a
larger regional scale, Kourtidis et al. (2002) used ozonesonde ascents, lidar observations, ship
cruises, and aircraft flights to show that south and southwestern synoptic flows associated with
Saharan dust events result in lower $O_3$ above the planetary boundary layer by 20–35 ppbv, as
compared to northerly flows, which transport air from continental Europe. Based on sixteen
years of $O_3$ concentrations measured at the EMEP Agia Marina Xyliatou rural background
station in Cyprus and 3 other remote marine sites, over the western, central and eastern parts of
the island, Kleanthous et al. (2014) have shown that local precursors contribute to only about 6%
(~3 ppbv) of the observed $O_3$. However, elevated concentrations of this secondary pollutant
occurring in summer are attributed to LRT of air masses mainly originating from northerly and
westerly directions. The summer average annual maximum of 54.3 ±4.7 ppbv was observed to be
related to the transport of polluted air masses from the Middle East, East and Central Europe
toward Cyprus.
Despite the prevailing synoptic meteorological conditions featuring the EM in summer, the
differing pathways of the LRT of polluted air masses can affect differently the buildup of
pollutants concentrations. To investigate such changes, Wanger et al. (2000) performed a
comprehensive study that included 150 hours of instrumented aircraft monitoring flights
comparing two events of air mass transport (September 1993 and June 1994) representing two
distinct types of LRT. This airborne study comprised flight paths performed approximately
70 km offshore parallel to the Israeli coastline and 180 km in length with Tel-Aviv in the center.
These flights were performed during midday under westerly wind flow conditions at an altitude
of 300 m a.s.l. (well within the atmospheric mixed layer). While both wind flow conditions were
nearly similar through the measurement periods and along the 180-km flight path, the air mass
sampled in September 1993 was much "cleaner" than the one sampled in June 1994. The
averaged $O_3$ concentration of the first campaign was 39 ±7 ppbv, against 48 ±9 ppbv in the
second period. Therefore, Wanger et al. (2000) model simulation showed that the pollution
sources in southern Europe and the Balkans did not affect the EM coasts in September 1993,
contrarily to the synoptic conditions and simulation results for the June 1994 period where the
winds over the EM tended to be northwesterly and thus forcing the polluted air masses toward
the coasts of the EM.
The summer synoptic and dynamic conditions prevailing over the EM supply the essential
ingredients for the building up of $O_3$ concentrations. Based on the similar climatic conditions
between the Los Angeles Basin (USA) and the EM, Dayan and Koch (1996) proposed a
theoretical description of the cyclic mechanism in summer, leading to fumigation (i.e., a
downward dispersion of an enriched $O_3$ cloud toward the ground) further inland from the EM
coast. Under the deep mode of the PT, stronger westerly winds, acting as a weak cold front (Fig.
15, panel A1), penetrate far inland, undercutting the mixed layer polluted by $O_3$ from the
previous day (Fig. 15, panel A2). In this way, part of the mixed layer containing $O_3$ is pushed
upward and isolated from the ground. If the pressure gradient weakens on the following day, the
western flow weakens (Fig. 15, panel B1). The cooling effect of the cool and moist marine air is
consequently reduced and the convective boundary layer inflates rapidly. When the top of the
mixed layer reaches the elevated $O_3$ cloud, the latter is penetrated by convective currents (Fig.
15, panel B2) and parts of the cloud are entrained toward the ground in this fumigating process.
Elevated $O_3$ concentrations (>117 ppbv) were measured at inland rural sites of central Israel
during the 1988-1991 early summer months (Peleg et al., 1994). Based on air mass back-
trajectory analyses, these elevated $O_3$ mixing ratios were found only in case of air masses
overpassing Tel Aviv metropolitan area. Furthermore, the very low ratio of $SO_2/NO_x$ (sulfur
dioxide, $SO_2$) clearly indicates that $O_3$ precursors such as NOx, CO, and VOC originate mainly
from fossil-fuel combustion from mobile sources (Nirel and Dayan, 2001). These pollutants are
subjected to chemical and photochemical transformations in the presence of solar radiation and
atmospheric free radicals to form $O_3$.
Over central Israel, the main source for these precursors emitted along the Israeli coastline is
transportation (Peleg et al., 1994). Since $O_3$ and other secondary pollutants formation takes
several hours, significant transport and mixing occur simultaneously with the chemical reactions
(Seinfeld, 1989; Kley, 1997). Thus, increasing urban and commercial activity along the highly
populated Israeli coastal region, together with expanding transportation activity in the Gaza
region, was found to strongly deteriorate inland air quality and, specifically, to cause
increasingly elevated inland $O_3$ levels. Model results showed that traffic emissions during the
morning rush hour from the Tel Aviv metropolitan area contribute about 60% to the observed $O_3$
concentrations (Ranmar et al., 2002). Moreover, their study showed the summer season features
a shallow mixed layer and weak zonal flow, leading to poor ventilation rates, which restrict $O_3$
dispersion efficiency. These poor ventilation rates result in the slow transport of $O_3$ precursors,
enabling their photochemical transformation under intense solar radiation during their travel
inland from the EM coast.
However, elevated $O_3$ concentrations are not limited to the summer over the EM. Dayan and
Levy (2002) found 103 ''high-ozone days'' where $O_3$ is >80 ppbv for at least 2 hours based on
24 Israeli sites over 1997-1999. From their $O_3$ temporal analyses, they concluded that the highest
values are more frequent during the transitional (spring and autumn) seasons (65% of 103 days)
than during the summer season (35%).
Based on the recent remote sensing tools in conjunction with meteorological observations and
models, we conclude on the three essential processes that control the $O_3$ concentration during
summer at various tropospheric levels over the EM: 1/ in the shallow troposphere, the horizontal
transport of $O_3$-enriched air masses from eastern continental Europe to the region controlled by
the anticylonic center over central and southeastern Europe and the PT causing the Etesians;  2/
the dynamic subsidence at mid-tropospheric levels; and 3/ the stratosphere-to-troposphere
exchange in the upper troposphere. At the surface of the EM coast, during transitional seasons,
high $O_3$ episodes are associated with hot and dry air masses originating east of Israel, where $O_3$
precursor emissions are negligible, demonstrating that high $O_3$ levels are more dependent on air
mass characteristics than on upwind precursor emissions.
**3.2 Particulate sulfate ($SO_4$) abundance**
Globally, the two-main particulate $SO_4$ precursors are $SO_2$ from anthropogenic sources and
volcanoes, and dimethyl sulfide (DMS) from biogenic sources, especially marine plankton. In
the EM atmosphere, particulate $SO_4$ contributes more than 50% to the submicron aerosol mass
(Bardouki et al., 2003a, b; Sciare et al., 2005). A first attempt to quantify the biogenic
contribution caused by the oxidation of marine DMS as possible source to particulate $SO_4$
observed over the EM coastal region, was carried out by Ganor et al. (2000). They used an
instrumented aircraft during August 1995 to sample DMS and methane sulphonic acid (MSA)

offshore and over land of Israel. Being exclusively produced by oxidation of DMS, MSA was used as tracer. Ganor et al. (2000) found this source as a rather limited contributor: between 6 and 22% of the non-sea-salt $SO_4$ (nss-$SO_4$) measured during summer was attributed to marine biogenic production. Evidently, several other factors favor particulate $SO_4$ abundance over the EM. The homogeneous conversion of gaseous $SO_2$ to particulate $SO_4$ is rather slow, i.e., about 1–3% per hour (Meagher et al., 1981). Wet deposition chiefly governs the atmospheric lifetime of $SO_4$, estimated to be up to 6 days on a global average (Chin et al., 2000). Due to rainless conditions and associated wet deposition in summer, and the slow dry deposition velocity of $SO_4$ aerosol (~0.01-0.4 cm s$^{-1}$), $SO_4$ aerosols account for 50-90% of the total sulfur (S) in transported air masses toward the EM (Matvev et al., 2002). Two additional factors favor late spring and summer particulate $SO_4$ regional abundance. First is the intense radiant energy emitted by the sun under clear sky conditions that leads to an efficient oxidation of $SO_2$ to $SO_4$ via hydroxyl radical (OH) as the predominant oxidant during daytime (Mihalopoulos et al., 2007). Second is the prevailing summertime westerly winds that transport $SO_4$-rich air masses from sources over central Europe before significant removal occurs. A pioneering study to measure particulate $SO_4$ in the background atmosphere of the EM was carried out by Mihalopoulos et al (1997) in Finokalia, Greece. They reported a mean $SO_4$ aerosol concentration of 188 neq m$^{-3}$ (~ 9 µg m$^{-3}$) with a minor marine contribution of about 5% resulting in a concentration of 178 neq m$^{-3}$ (~8.5 µg m$^{-3}$) for nss-$SO_4$. These summer concentrations, about 10% higher than those observed in Thessanoliki (Tsidouridou and Samara, 1993), were associated with transport from eastern and central Europe. This is consistent with Sciare et al. (2003) who measured particulate nss-$SO_4$ during a one-month experiment in summer 2000 at a background site on Crete. They found a high average concentration of 6 µg m$^{-3}$ (~62 nmole m$^{-3}$) for air masses originating from Turkey and Central Europe. Identical results were obtained by Koulouri (2008) who measured similar nss-$SO_4$ concentrations during the period July 2004–July 2006.

Another source of $SO_4$ aerosols is ship emissions, which contribute substantially to atmospheric pollution over the summertime Mediterranean region. Based on a regional atmospheric-chemistry model and a radiation model, Marmer and Langmann (2005) found that the summer mean $SO_4$ aerosol column burden over the Mediterranean is 7.8 mg m$^{-2}$, 54% originating from ship emissions.

Concentrations of $SO_4$-rich air masses have been measured intermittently at various downwind
ground sites in Israel, the easternmost Mediterranean region, from an instrumented aircraft for a
10-year period between 1984 and 1993 by Luria et al. (1996). They found that the concentration
of particulate $SO_4$ observed during the summer was relatively high compared to other world
locations, exceeding occasionally 500 nmole m$^{-3}$ as compared to wintertime levels that were in
the range of 50-100 nmole m$^{-3}$. From airborne observations, Wanger et al. (2000) measured an
averaged $SO_4$ concentration of 38 ±7 nmole m$^{-3}$ in their first series of measurement between 5
and 9 September 1993, and up to 108 ±63 nmole m$^{-3}$ between 15 and 21 June 1994. The annual
average, calculated in Luria et al. (1996), is 100 ±15 nmole m$^{-3}$, which is twice as high as
predicted for the region by a global model and as high as reported for some of the most polluted
regions in the USA. They pointed to several indicators suggesting that the origin of the
particulate $SO_4$ over the EM region is not from local sources but the result of LRT. The
indicators include the lack of correlation between $SO_4$ and primary pollutants, the high $SO_4$ to
total S values, the origin of the air mass back trajectories and the fact that similar levels were
observed during concurrent periods at different sites. Throughout their study, a higher
concentration of $SO_4$ was found during the afternoon hours, especially during the summer and at
the inland locations. However, aerosol chemical analyses from a two-stage aerosol sampler from
a receptor site in Sde-Boker (31.13°N, 34.88°E, 400 m a.s.l.) in southern Israel, point at a
significant decline of 24% of these elevated nss-$SO_4$ mean concentrations for the summer months
(July and August) from ~3 µg m$^{-3}$ in 1994 to ~2.3 µg m$^{-3}$ for 2004. This decline is attributed to
the decrease of S emissions in central and eastern Europe over the past 3 decades. Indeed, the
majority (60%) of the calculated air mass back trajectories related to extreme events (during
which the fine fraction S concentration at Sde-Boker exceeded a threshold of 3 µg m$^{-3}$)
originated from Russia, Ukraine and northern Black Sea region (Karnieli et al., 2009).
The effect of land and sea breeze on coastal meteorology in general and the interaction between
land and sea breeze and air pollutants in particular plays an important role in determining many
aspects of coastal environments around the world. A meteorological phenomenon that is often
associated with the land and sea breeze is air mass recirculation in coastal regions (Miller et al.,
2003; Levy et al., 2008). Sulfate particles measured along the central coast of Israel in mid-
August 1987 and mid-August 1995 and identified by lesser microprobe analysis have shown that
the concentration during land breeze were 6-10 times higher (34.6–64.1 µg m$^{-3}$) as compared to
sea breeze conditions (4.3–7.1 µg m$^{-3}$) (Ganor et al., 1998).
In another attempt to quantify the S flux arriving at Israel's western coast from Europe and the
Israeli pollution contribution to the air masses leaving its eastern borders towards Jordan, Matvev
et al. (2002) conducted 14 research flights at an altitude of approximately 300 m above ground
level, measuring $SO_2$ and particulate $SO_4$ during the summer and autumn seasons. Two different
legs were performed for each research flight: the first over the Mediterranean Sea, west of the
Israeli coast and the second along the Jordan Valley. Their results have shown that the influx of
S reaching the Israeli coast from Europe varied in the range of 1–30 mg S h$^{-1}$, depending on the
measuring season. The $SO_4$ level in the incoming LRT air masses was at least 50% of the total S
content. The contribution of the local pollutant sources to the outgoing easterly fluxes also
strongly varied with the season. The Israeli sources contributed an average of 25 mg S h$^{-1}$ to the
total pollution flux during the early and late summer as compared to only approximately
9 mg S h$^{-1}$ during the autumn period. The synoptic analysis indicates that conditions during the
summer in Israel favor the accumulation of pollution species above the Mediterranean Basin
from upwind European sources. This season is characterized by weak zonal flow within a
shallow mixed layer that lead to poor ventilation rates, limiting an efficient dispersion of these
pollutants during their transport eastward. Under these summer conditions, in-flux local
contribution and the total out-flux of these pollutants are elevated as opposed to other seasons.
To illustrate, during autumn, the EM is usually subjected to weak easterly winds, interrupted at
times by strong westerly wind flows inducing higher ventilation rates. Such autumnal
meteorological conditions and the lack of major emitting sources eastwards of Israel result in
lower S budgets to and from Israel.
An estimate of the yearly flux showed that approximately 0.06 Tg S arrived at the Israeli coast
from the west (Matvev et al., 2002). This is approximately 15% of the pollution leaving Europe
towards the EM. The outgoing flux towards Jordan contributed by local sources was calculated
to be 0.13 Tg S per year, i.e. almost all the S air pollution emitted in Israel. The results of the
flux rates for the S compounds over Israel are summarized in Table 2 for the different research
flights and field campaigns. These latter results show for the early summer time that the
uppermost fluxes from the west were averaging 0.19 Tg y$^{-1}$. During this season, the levels
doubled the averages for late summer (0.085 Tg y$^{-1}$) and were over five times the average levels
measured for the autumn (0.035 Tg y$^{-1}$). The wide range in fluxes derived is explained by the
varying distance from the polluted coastline.
The Aerosol Optical Depth (AOD), the vertical integral over an atmospheric column of the
incident light scattered and absorbed by aerosols, is often used to estimate the aerosol loading in
the atmosphere. Particulate SO$_4$ are among the numerous aerosol types. Nabat et al. (2013)
compared AOD from several model data to satellite derived data for the period 2003-2010 over
the Mediterranean region. They found that the AOD seasonal cycle obtained from the
Monitoring Atmospheric Composition and Climate (MACC) reanalysis model, which includes
Moderate-Resolution Imaging Spectroradiometer (MODIS) AOD assimilation at 550 nm
resembles much the satellite-derived AOD variability and have the best spatio-temporal
correlation compared to AErosol RObotic NETwork (AERONET) stations. Based on these
models and satellite derived data, Nabat et al. (2013) have clearly shown that particulate SO$_4$,
has a maximum during spring and summer over the EM (Fig. 16). Matvev et al. (2002)
performed airborne measurements along a 150-km line west of the Israeli coast. They derived an
annual flux of the order of 0.06 Tg yr$^{-1}$ of (dry) S across the corresponding surface. Given the
observed ratio of SO$_4$ to total S of 40-90% in the region (Matvev et al., 2002; Sciare et al., 2003),
the annual flux of SO$_4$ based on field measurements is 0.024-0.054 Tg y$^{-1}$. Rudich et al. (2008)
used satellite data to estimate the pollution transport toward the EM. MODIS Terra- and Aqua-
derived estimates of the annual SO$_4$ flux along the same transect are 0.038 and 0.040 Tg y$^{-1}$,
respectively, in the middle of the range obtained from field observations.
Rudich et al. (2008) also found that MODIS-based estimates (from Terra and Aqua satellites) of
the SO$_4$ flux agree reasonably well with the Goddard Chemistry Aerosol Radiation and Transport
(GOCART) model simulations of anthropogenic SO$_4$, as shown in Figure 17 for seasonal
averages. The annual SO$_4$ flux from the GOCART model is 0.181 Tg y$^{-1}$, about 18% higher than
the MODIS/Terra estimate of 0.153 Tg y$^{-1}$. Similar comparison on a seasonal basis exhibits that
GOCART model over estimates the winter (by ~85%) and spring (by ~30%) fluxes whilst lower
estimates the summer and autumn fluxes by 10-25%. If we consider the comparison between the
GOCART model and MODIS/Aqua, the model annual flux is 0.201 Tg y$^{-1}$ , about 25% higher
than the MODIS/Aqua estimate of 0.159 Tg y$^{-1}$. On a seasonal basis, their estimates are in
excellent agreement in summer and fall, but about 50% higher in the MODIS/Aqua winter and
spring estimates. Based on the comparison of the two instruments, the model results, and the
consistency with the aircraft measurements, they concluded that both MODIS instruments can be
used for estimating the flux of pollution based on their daily AOD retrievals.
**3.3 Local formation and long-range transport of total reactive nitrogen ($NO_y$)**
Total reactive nitrogen ($NO_y$) is a collective term for oxidized forms of nitrogen in the
atmosphere such as nitric oxide (NO), nitrogen dioxide ($NO_2$), nitric acid ($HNO_3$), nitrous acid
($HNO_2$), nitrate ($NO_3$), nitrogen pentoxide ($2N_2O_5$), peroxynitric acid ($HNO_4$), peroxyacetyl
nitrate (PAN), and other organic nitrates (Emmons et al., 1997). Research studies measuring
inorganic reactive nitrogen compounds over marine areas in general, and more specifically over
the EM basin are scarce (Lawrence and Crutzen, 1999; Corbett et al., 1999; Veceras et al., 2008).
Measurements of $NO_2$, $HNO_3$ and $HNO_2$ undertaken with instrumentation on board a research
vessel in the Aegean Sea between 25 to 29 July 2000 revealed typical $NO_2$ concentrations of 4–
6 ppbv with a broad maximum of 20–30 ppbv. The level of $NO_2$ was relatively high during the
night and low during the day due to enhanced photochemical activity, vertical mixing and the
daily wind characteristics. Extreme $NO_2$ concentration were caused by up slope wind bringing
air from marine traffic emissions trapped within the marine atmospheric boundary layer. The
concentration of both, nitric and nitrous acids, in ambient air of the Aegean Sea was low, below
50 pptv. Večeřa et al. (2008) explained these results by the lack of precursors for these acids
(Cohen et al., 2000), the high solar irradiation leading to $HNO_3$ dissociation, and the reaction of
$HNO_3$ with sodium chloride aerosol.
$NO_y$, identified as precursors in the $O_3$ formation, was measured by Wanger et al. (2000) for two
summer airborne campaigns over the EM at an altitude of about 300 m (well within the MLD)
using a high-sensitivity NO-$NO_y$ analyzer (TEII 42 S, chemiluminescence method, ±0.1 ppbv
sensitivity). In the first campaign of September 1993, characterized by cleaner air mass
conditions, an average $NO_y$ concentration of 1.0 ±0.6 ppbv was measured as compared to 3.9
±1.8 ppbv sampled during the June 1994 campaign.
The Mediterranean Intensive Oxidant Study (MINOS) campaign, performed in the summer of
2001, allowed Lelieveld et al. (2002) to examine the air pollution conditions at shallow and mid-

tropospheric levels over the EM Basin. During this experiment, elevated concentrations, typically 0.1 to 0.2 ppbv, of NO in the upper troposphere and only about 20 pptv within the MLD were observed at the Finokalia station. However, the value measured within the MLD at Finokalia was rather low and not typical for this site. From fall 1998 to summer 2000, a Thermo Environmental Model 42C high-sensitivity chemiluminescence $NO_x$ analyzer with a detection limit of 50 pptv was operated at Finokalia in parallel with the $O_3$ analyzer to monitor NO and $NO_x$ (Kouvarakis et al., 2002). During the whole examined period, NO concentrations ranged between 50 pptv (most of the time) and 100 pptv, and $NO_x$' ($NO_x$' = NO + $NO_2$ + PAN) between 0.1 and 4 ppbv. Kouvarakis et al. (2002) interpreted the very low NO/$NO_x$' ratio obtained as pointing at the influence of the Finokalia station by aged air masses. Furthermore, they argued that the similar diurnal amplitude of $O_3$ above the Aegean Sea and at Finokalia during summer, indicates that the regime of $NO_x$ above the Aegean is similar to that observed at Finokalia.

The observed diurnal evolution at Finokalia of NO and $NO_z$' - the later expressing mainly the sum of $NO_2$, NO, PAN-like compounds, organic nitrates and $HNO_3$ - were used as tracers of pollution by Gerasopoulos et al. (2006) to analyze the diurnal variability of $O_3$ over the EM. The diurnal cycles of these two tracers based on 3.5 year of measurements point at a maximum value of ~70 pptv for NO and up to ~1.55 ppbv for $NO_z$'. These maxima were observed 1-2 h after the minimal $O_3$ concentration measured at about 06:30 UTC.

Ambient concentrations of NO, $NO_2$ and $NO_x$ have been also reported over the northwestern parts of Turkey. An $NO_2$ concentration of 8.5 ±4.8 ppbv was obtained for the summer of 2005 by collecting weekly average data in a sampling site of the city Eskişehir, located 230 km to the west to the capital of Turkey by use of passive samplers (Ozden et al., 2008). Im et al (2008) studied $O_3$ pollution and its relationship with $NO_x$ species based on hourly concentration levels of $O_3$, NO, and hydrocarbon measured between 2001 and 2005 in Kadıköy, an urban district in the Anatolian side of İstanbul. The mean and standard deviation for the summer (June-August) NO, $NO_2$ and $NO_x$ concentrations reported for this 5-yr period were 14.4 ±6.2, 22.75 ±2.7, and 37.7 ±14.3 ppbv respectively. Moreover, they suggested that the very strong correlation they found between NO and $NO_x$, implies that the $NO_x$ species are mainly from local sources.

Traub et al (2003) analyzed several trace gas concentrations measured along flight tracks of the Deutsches Zentrum für Luft- und Raumfahrt (DLR) Falcon aircraft over the eastern and central

Mediterranean Sea during MINOS in August 2001. In order to inquire into the role of LRT of
pollutants in the air masses above the Mediterranean area and to determine their source regions,
5-day backward trajectories were computed and initialized along the Falcon flight tracks. They
found that all trajectories with source regions in eastern Europe were associated with higher
mean concentrations than those from westerly directions. Traub et al (2003) measured a mean
NO and NOy concentration of 0.05 ±0.02 and 1.4 ±0.4 ppbv respectively for the computed
trajectories within the MLD originating from eastern Europe as compared to 0.04±0.01 and
1.1±0.5 ppbv respectively for trajectories originating from western Europe. .
Increasing urban and commercial activity along the highly populated Israeli coastal region,
together with expanding transportation activity has yielded few ground-based measurements
studies in order to quantify the impact of local urban versus regional and foreign sources on the
concentrations of the $NO_x$ species, which vary in their atmospheric fate.
Results of half-hourly $NO_x$ concentrations recorded from 9 monitoring stations from 2002 to
2005 in the Haifa Bay, Israel, resulted in a typical mean mixing ratio of 25 ppbv (Yuval et al.,
2007) and a typical background value below 0.5 ppbv for the summer over the EM (Alpert-
Siman Tov et al., 1997). This background value was further evidenced by Dayan et al. (2011)
who analyzed $NO_x$ concentrations during the Day of Atonement. In this day, all traffic and most
of the industrial activities cease in the Jewish populated parts of the country, which provides a
unique opportunity to test the relative contribution of pollution sources within urban centers
versus regional and foreign sources.
In a study aimed at analyzing the sources and sinks of HONO in urban areas, and their seasonal
dependency, Amaroso et al. (2008) carried out measurements of HONO, $NO_x$, $O_3$, and $SO_2$
during autumn and summer in Ashdod (31°49′N, 34°40′E, 10 m a.s.l.) (south of Tel Aviv,
Israel), a typical coastal Mediterranean urban area. The 15-day July campaign consisted of 5-min
averaged 4320 measurements, of HONO, NO and $NO_2$. HONO analyses were performed with a
liquid coil scrubbing/UV-vis instrument (see Amaroso et al., 2008). NO and $NO_2$ measurements
were performed by a thermos Model 42C NO-NOx analyzer. The mean concentration obtained
for this campaign was 1.4 ±2.0, 6.0 ±8.8 and 14.8 ±7.3 ppbv for HONO, NO, and $NO_2$,
respectively. The HONO mixing ratios obtained clearly point at the typical diurnal cycle with
nighttime maxima and daytime minima (Lammel and Cape, 1996).
Ranmar et al. (2002) addressed the dynamics of transboundary air pollution, where transportation
emissions (such as $NO_x$ and VOC) originating from Israeli major coastal sources impact the
onshore mixing layer. Analysis of $NO_y$ data (here, the sum of all nitrogen oxide species,
excluding $N_2O$) collected from 1 June to 30 September for the years 1999 and 2000 at a
monitoring station located in metropolitan Tel Aviv, yielded an average of 24.5 ±15.1 ppbv.
They noted the higher initial $NO_y$ levels during the morning rush hour emissions that were
subjected to a noticeable bleaching by the late morning sea breeze in comparison to inland
locations, which leveled off at relatively higher midday concentrations. Ranmar et al. (2002)
argued that this may indicate, in the absence of any alternative $NO_y$ source, that the early
morning $NO_x$ produced by transportation sources in Tel Aviv is transported inland, providing
additional $NO_y$ to the regions along its path.

Beside cruises of research vessels, airborne campaigns, and ground truth measurements, satellite-
borne initiatives have been undertaken to get a better insight on the reactive nitrogen
concentrations over the EM. Marmer et al. (2009) used OMI (Boersma et al., 2007) as an
observation tool to measure atmospheric $NO_2$ column concentrations in order to validate ship
emission inventories over the Mediterranean Basin. Figure 18 shows the average OMI $NO_2$
tropospheric columns (gridded to 0.125°×0.125°) over the Mediterranean Sea for June-August
2006. The most prominent feature here is the elevated $NO_2$ monthly mean. Under cloud free
conditions, typical values ranged from 1.2 to 2.0 $10^{15}$ molecules $cm^{-2}$ over the northeastern
African coast, the EM coast, the southern coast of Turkey and the whole Aegean Sea, as
compared to over 6 $10^{15}$ molecules $cm^{-2}$ for European inland congested regions. Based on OMI
$NO_2$ tropospheric columns and the Goddard Earth Observing System chemistry transport
(GEOS-Chem) model, Vinken et al. (2014) attributed the elevated $NO_2$ columns regions over the
Mediterranean to $NO_2$ emissions along ship tracks.
**3.4 Carbon Monoxide sources and pathways**
CO has a global-average lifetime of about two months in the troposphere and its molecular
weight is close to that of air. This molecule is considered as an excellent tracer for pollution
sources and pollution pathways through the troposphere. In addition to production by chemical
oxidation in the atmosphere, CO is emitted by biomass burning, man-made sources, vegetation,
and ocean. The CO seasonal cycle is mainly governed by the concentration of OH in the
troposphere (Novelli et al., 1992) and is expected to be the lowest in the summer when
photochemistry is active and the highest during late winter or spring.
An assessment of CO baseline concentration levels at the surface over the EM is presented based
on few observational studies that have been conducted for this pollutant. As part of a
comparative air quality study, CO was analyzed at Patras (38.25°N, 21.74°E) and Volos
(39.36°N, 22.94°E), two Mediterranean Greek coastal urban sites (Riga-Karandinos and Saitanis,
2005). They observed an annual average hourly mean concentration of 1.14 ppm over 1995-2003
at Volos as compared to 0.95 ppm at Patras over 2001-2003. The diurnal pattern at both sites
during summer showed that vehicle-induced emissions contribute significantly to CO levels with
peak concentrations of 1.14 and 0.96 ppm measured at 09:00 UTC at Volos and Patras,
respectively. Over the EM coast, hourly average CO measurements conducted by Saliba et al.
(2006) in the city of Beirut (33.89°N, 35.50°E), Lebanon, point at an average monthly CO
concentration during summer of 1.05 ppm, similar to the concentrations observed in Volos and
Patras, Greece (Riga-Karandinos and Saitanis, 2005).
CO concentrations were measured by Elbayoumi et al. (2014) from the fall of 2011 through mid-
2012 in the Gaza strip, in the southeastern coast of the EM as part of an exposure study to assess
the effect of seasonal variation on the mean daily indoor-outdoor ratio at 12 schools located over
the northern, central and southern strip of Gaza. They observed a six–hour average daily outdoor
CO concentrations of 0.96 ±0.91 ppm for all the schools. They further reported that the outdoor
CO concentration spanned from 0.10 ppm to 2.46 ppm with a mean of 0.88 ppm for urban sites
and from 0.10 to 2.71 ppm with a mean of 1.02 ppm for overpopulated sites along the Gaza strip.
Due to the key role CO plays in atmospheric chemistry, several chemistry-transport modeling
studies were devoted to this subject. CO was measured and used as a tracer in such a model
(Lelieveld and Dentener, 2000) during the summer 2001 MINOS campaign (Lelieveld et al.,
2002). The model diagnosed CO from anthropogenic sources in different parts of Europe, North
America, and Asia. Trajectory calculations in the lower troposphere identified western and
eastern Europe as the main source emissions. Consequently, model simulations were performed
for August 2001 over Sardinia (40°N, 8°E) in the western Mediterranean and over Crete (35°N,
25°E). Considering the negligible impact of local pollution sources, the high CO levels observed
over Crete, in excess of 150 ppbv, were surprising. The model results indicated that regions
surrounding the Mediterranean such as southern Italy, Greece, Serbia, Macedonia, the Middle
East, and North Africa contribute relatively little to the CO pollution, typically about 20%.
Furthermore, Lelieveld et al. (2002) found that the EM is affected by CO polluted air emitted
from eastern Europe, Poland, the Ukraine, and Russia. This pollution flow, east of the Carpathian
Mountains, is channeled over the Black Sea and the Aegean Sea, and contributes 60 to 80% of
the boundary-layer CO over the EM. Their model results are consistent with aircraft
measurements, showing that the entire Mediterranean lower troposphere is polluted.
In the free EM troposphere, where westerly winds predominate, they revealed a quite different
situation as compared to concentrations measured within the MLD. The mid-tropospheric CO
measurements were ~75-80 ppbv. From their model tracer analysis, the largest contribution over
the Mediterranean is found originating from Asia (40 to 50%). The CO typical lifetime
(~2 months) enables air mass to circumnavigate the globe, which results in a low variability of
its concentrations. Lelieveld et al. (2002) found that contributions by pollution from western and
eastern Europe to mid-tropospheric CO were only about 10%.
Drori et al. (2012) conducted a study to locate the various CO sources converging from Europe,
North Africa and the Middle East and quantify their respective contributions to the EM.
Background CO concentrations are monitored regularly over the southern part of Israel in Sde-
Boker (Weizmann Institute of Science – WIS Station Negev Desert: 31.13°N, 34.88°E, 400 m) as
part of the National Oceanic and Atmospheric Administration (NOAA) Earth System Research
Laboratory Global Monitoring Division (ESRL/GMD), which aims at representing the EM.
While comparing the seasonal cycle of Sde-Boker to other European ESRL/GMD background
sites (see Table 3), one essential feature is eminent from their results represented in Figure 19:
CO concentrations are high over winter months, decreasing abruptly during April and increasing
again from November. A second maximum is observed during August compared to July and
September (Drori et al., 2012).
To get an insight on the spatial distribution of CO concentrations over the EM, the Version 4
Measurement of Pollution in the Troposphere (MOPITT) level-2 CO retrievals (Deeter et al.,
2010) were employed by Drori et al. (2012) using a priori information for MOPITT V4 CO
retrievals based on the Model for OZone and Related chemical Tracers (MOZART-4) chemistry-
transport model simulation climatology (Emmons et al., 2010). The averaging kernel profile
obtained for a retrieval near Sde-Boker ESRL/GMD station shows that, during the day, the
900 hPa retrieval sharply peaks at the same level, indicating that there is a good sensitivity to
lower tropospheric concentration. The anomalous high concentration observed at the WIS
ESRL/GMD Sde-Boker station, and calculated by the MOZART-4 model during August (Fig.
19), might be limited to lower levels, and therefore averaging over several layers might hide this
signal. Furthermore, Drori et al. (2012) compared the in-situ measurements at Sde-Boker and CO
retrieved from MOPITT to MOZART-4 model results. CO sources included direct emissions and
secondary production from hydrocarbons oxidation, while CO sinks included a reaction with OH
and dry deposition. The seasonal cycle of surface CO at Sde-Boker simulated by MOZART and
averaged for five consecutive years shows a similar pattern exhibiting CO concentration reaching
a maximum in February and a second peak in mid-summer months (i.e., July and August) that
surpasses those of the early summer (i.e., May–June) (Fig. 19).
To attribute the CO sources affecting the EM, Drori et al. (2012) partitioned these sources using
a tagging method into five types: anthropogenic, biogenic, fire, chemical production, and ocean.
The total CO concentration and specific contributions 2006–2007 times-series of MOZART at
the surface at 30° N and 33.75° E are shown in Figure 20 where ocean sources contributions are
not shown (negligible). Both biogenic (green line) and biomass burning sources (red line) have a
minor contribution. Biogenic sources are characterized by a distinct seasonal cycle with high
contribution over winter and low daily variability. Biomass burning has no defined seasonal
signature and contributes on an episodic event basis. CO from chemical production (orange)
contributes substantially (50–80 ppbv) with a defined seasonal cycle: low during winter and
autumn and high during summer featured by a low daily variability. Anthropogenic sources were
found to be the main contributor to the total CO (purple, 50–180 ppbv). As expected, their
seasonal cycle is featured by winter elevated concentrations decreasing during spring, slightly
increasing during summer and decreasing again during autumn. The daily variability is high and
similar to the total CO daily variability. Comparing the daily variability of the various sources,
Drori et al. (2012) concluded that anthropogenic sources mainly govern total CO daily variability
over the EM.
To further attribute the CO surface daily variation, Drori et al. (2012) tagged the anthropogenic
sources for the three northern continents, i.e., North America, Europe, and Asia. Figure 21 shows
the results of these anthropogenic sources attribution to the CO surface. European anthropogenic
sources contribute substantially (10–80 ppbv) to local CO concentrations with the greatest daily
variability all year round. Asian and North American sources are in the same order of magnitude
(10–25 ppbv) with low daily variability during most of the year, and very small variability during
summer. Obviously, daily summer CO variations in the EM are mainly caused by European
anthropogenic sources. The seasonal cycle of the European contribution is very similar to the
seasonal cycle of total CO, featured by a high concentration in winter, spring, and autumn and a
lower summer concentration. The contribution of European emissions to CO surface
concentrations is comparable to that from EM local emissions.
Drori et al. (2012) found, however, that local and European emission contributions to local CO
concentrations are generally negatively correlated, meaning that either local or European sources
are dominant, except during summer, when both sources affect simultaneously the local CO
concentration. A possible explanation for the positive summer correlation might be explained by
the short range of air mass transport caused by the dominant summer synoptic system, i.e., the
PT in its weak mode recirculating local and European emissions, and by the fact that summer
chemical production is a major CO source over the EM.
Another recent modeling study focused on CO concentrations was conducted by
Myriokefalitakis et al. (2016). They compared and validated model results against in-situ
observations at the surface, in the mixed layer and in the free troposphere (between 850 hPa and
the tropopause) in the countryside and remote atmosphere over Europe for 2008. This study
analyzes the total CO budget and the partial contribution of regional anthropogenic, biogenic and
biomass burning CO emissions in the EM. The budget calculated for 2008 in the EM mixed
layer, using a basic simulation relying on anthropogenic emissions and meteorology, points at a
load of 0.6 Tg of CO, a chemical production of 10 Tg yr$^{-1}$, primary emissions in the region of
8 Tg yr$^{-1}$ and a dry deposition flux of 3 Tg yr$^{-1}$. Moreover, Myriokefalitakis et al. (2016) found
that subsidence from higher atmospheric layers typifying the EM summer is an important CO
source (12 Tg yr$^{-1}$) in the EM free troposphere. At the surface, anthropogenic local emissions in
the EM were found to contribute by 18% to surface CO levels on an annual average. Over Cairo,
out of the total surface CO concentration, roughly 32% are contributed by anthropogenic
sources. These EM CO concentration results are consistent with previous modelling studies (e.g.,
Kanakidou et al., 2011; Drori et al., 2012; Im and Kanakidou, 2012).

**3.5 Methane concentrations**

$CH_4$ is the most abundant hydrocarbon in the atmosphere with concentration originating from
natural and anthropogenic sources. It is also the most contributor to GHG after water vapor and
$CO_2$ due to its high global warming potential relying on its infrared absorption and long
atmospheric lifetime of ~8 years (Lelieveld et al., 1998), which allows its mixing throughout the
atmosphere. $CH_4$ emissions are primarily caused by microbiological decay of organic matter
under depletion of dissolved oxygen in wetlands, followed by decomposition of solid waste and
enteric fermentation from domestic livestock. As for the geologic sources, a total geological $CH_4$
flux of $53 \pm 11$ Tg $yr^{-1}$ was suggested, which accounts for 7–10% of the total global $CH_4$ budget
(Etiope et al., 2008). The geological formations contributing to $CH_4$ over the greater area of the
EM (25-50°N, 5°-55°E) are mud volcanoes with essential hot spots located over eastern
Romania, the Black Sea, central and eastern Azerbaijan, and the Caspian Sea.
In contrast to trace gases of short lifetimes such as NOx and NOy, the long lifetime of $CH_4$ over
the EM may lead to interannual fluctuations of concentrations caused by circumglobal
phenomena such as low frequency global circulation patterns, i.e., El Niño-Southern Oscillation
(ENSO) and North Atlantic Oscillation (NAO), or changes in global temperature. Langenfelds et
al. (2002) point at major biomass burning events linked to ENSO dry periods, which increased
the growth rate of $CH_4$ over other parts of the world. Artuso et al. (2007) compared the global
average temperature anomaly to the growth rate of $CH_4$ in Lampedusa (35.5°N, 12.6°E) Italy for
the period 1995-2005. The 0.71 positive correlation they found reflects the strong relationship
between these two factors. Over the EM, the NAO may possibly affect the concentration
evolution through changes in the circulation (e.g., weakening of the northwesterly flow).
However, so far, no association was found between the NAO index trend and the $CH_4$
concentration growth over this part of the basin. The only study analyzing directly a possible
association between the NAO index and $CH_4$ concentration growth carried out by Chamard et al.
(2003) in Lampedusa have not found any relationship between these two factors.
Satellite ability to monitor the concentration of trace gases in the atmosphere is important for
completing the picture as regarded to their budget. Among the space-borne measurements of
trace gases, the Scanning Imaging Absorption Spectrometer for Atmospheric Cartography
(SCIAMACHY) instrument was proven as a feasible tool to detect $CH_4$ concentrations
(Bovensmann et al., 1999). Measurements of column-average volume mixing ratios of $CH_4$ were
retrieved on a global basis (Frankerberg et al., 2005).
Georgoulias et al. (2011) used data from the SCIAMACHY instrument on board the European
environmental satellite (ENVISAT). SCIAMACHY's spectral near-infrared nadir measurements
are sensitive to $CH_4$ and $CO_2$ concentration changes at all atmospheric altitudes, including the
one in the mixed layer where the signal emitted from the surface source is the largest. Annual,
seasonal and monthly spatial distribution of $CH_4$ were displayed for 2003 and 2004 based on the
analysis of Weighting Function Modified Differential Optical Absorption Spectroscopy (WFM-
DOAS) version 1.0 (Schneising et al., 2009) dry air column-averaged mole fractions, denoted as
$XCH_4$ (ppbv). The reflectivity of water surfaces is very low, therefore Georgoulias et al. (2011)
mapped the concentration of $CH_4$ over the EM Basin discarding the Mediterranean Sea. To
reduce the noise inserted by the single pixel retrieval error and the temporal and spatial sparsity
of the data, the data were averaged on $1° \times 1°$ monthly mean grids. Annual, summer and August
spatial distributions for 2003 are displayed on Fig. 22 top, mid and bottom panel, respectively.
Those maps illustrate an eminent seasonal variation with a summer maximum in $XCH_4$ levels
observed in both consecutive years (2004 not shown). The northeastern African coast exhibits
the highest $XCH_4$ values, with a hot spot over the Nile's delta in Egypt in summer and August.
The lowest $XCH_4$ levels along the Arabian Peninsula, the Zagros Mountain and eastern Anatolia
mountain barrier coincide spatially with high altitude areas. To examine to what extent the warm
period affects the annual, seasonal, and latitudinal patterns, Georgoulias et al. (2011) further
proceeded to a monthly analysis. They observed an increase in $XCH_4$ levels during the summer
season, August being the month with the highest levels of 1775-1780 ±24 ppbv for both 2003
and 2004. The highest values are concentrated in the northeastern part of the area primarily in
July-August. From July to September, there is a shift of high $XCH_4$ levels from higher to lower
latitudes. Despite the abundance of mud volcanoes over the Greater Area of the EM region,
Georgoulias et al. (2011) ruled out the possibility that the $CH_4$ total columns from
SCIAMACHY (2003-2004) measured over these EM regions were attributed to volcano
eruptions.
Ricaud et al. (2014) presented a thorough analysis of atmospheric $CH_4$ distributions over the
Mediterranean Basin in the troposphere, as part of the Chemical and Aerosol Mediterranean
Experiment (ChArMEx) program, using both satellite measurements and model simulations. For
this sake, they analyzed space-borne measurements from (i) the Thermal And Near infrared
Sensor for carbon Observations-Fourier Transform Spectrometer (TANSO-FTS) instrument on
the Greenhouse gases Observing SATellite (GOSAT) satellite, (ii) the Atmospheric InfraRed
Spectrometer (AIRS) on the AURA platform and (iii) the Infrared Atmospheric Sounder
Interferometer (IASI) instrument aboard the MetOp-A platform. These space-borne tools were
used in conjunction with the results obtained from three global models: the chemical transport
model (CTM) MOCAGE (Teyssedre et al., 2007), and the two chemical climate models (CCMs)
CNRM-AOCCM (Michou et al., 2011) and LMDz-OR-INCA (Hourdin et al., 2006). The
sensitivity of those space-borne sensors is mainly located in the upper tropospheric layers
peaking around 300 hPa with an envelope as defined by the half-width at half-maximum of the
averaging kernels (see Figure 23) from 400 to 200 hPa. Consequently, the comparisons between
measurements and model outputs of $CH_4$ is mainly concentrated on the layer around 300 hPa for
AIRS and GOSAT, or considering the total column for IASI.
In summer, the horizontal distribution of $CH_4$ in the upper troposphere shows a clear longitudinal
gradient between the East and the West of the Mediterranean Basin, both in the space-borne
measurements and in the model calculations (Figure 24). There is a maximum of $CH_4$ in the
eastern MB compared to the western MB, both considering the upper tropospheric layer and the
total column information. The difference between the East and the West of the MB has been
calculated within all the datasets and the seasonal variations has been investigated (Figure 25).
This clearly shows that the East-West difference peaks in summer, mainly in August.
The LRT conditions in the upper troposphere differ over both parts of the Mediterranean Basin.
In the western part, whatever the season considered, air masses are basically coming from the
west. However, in the EM, apart from the westerlies influence, air masses are also originating
from northern Africa and the Arabic Peninsula (Ziv et al., 2004; Liu et al., 2009), and even
farther away, from Asia.
To further examine the origin of air masses reaching the eastern MB, a six-day back-trajectory
from the point at 33°N, 35°E located in the EM (red filled circle in Fig. 26) was calculated,
considering vertical movement, using the British Atmospheric Data Centre (BADC) trajectory
service (http://badc.nerc.ac.uk/community/trajectory/) every 12 h in July-August over 2001-
2010. The position of the gravity center of all trajectories (i.e. the maximum in the probability
density function) is displayed every 24 h in Figure 26 at 850 (red stars), 700 (orange), 500
(green), 300 (blue) and 200 hPa (yellow). For this purpose, data from ECMWF archive (2.5
degree/pressure levels) were used in the calculation.
Based on these studies focused on the EM, Ricaud et al. (2014) proposed a scheme displaying
the transport mechanism (Fig. 27) representing the several stages process: (1) capturing of lower
tropospheric pollutants, including $CH_4$, in the Asian monsoon; (2) pollutants ascent to the upper
troposphere by the Asian monsoon ; (3) accumulation of pollutants within the Asian monsoon in
the upper troposphere; (4) long-range transport and large-scale repartition of pollutants in the
upper troposphere from the Asian monsoon anticyclone to the Middle East and North Africa; (5)
subsiding air masses yielding to the build-up of pollutants at mid-tropospheric layers above the
EM.

## 4. Conclusions and perspectives

This review demonstrates the significant progress made in understanding the atmospheric
pollution over the MB. Measurements from space-borne and aircraft instruments and outputs
from chemistry-climate models and chemistry transport models clearly revealed that the general
atmospheric dynamic summer conditions characterizing the EM basin differ much from the
western ones. The impact of the different meteorological regimes together with the seasonal
variabilities of the emissions of various atmospheric pollutants result in a longitudinal
concentration gradient between the eastern and the western Mediterranean Basins.
Several new campaigns have been recently organized to give more insights in the understanding
of the processes occurring in the western and eastern parts of this basin in the framework of the
ChArMEx program. The TRAnsport and Air Quality (TRAQA) campaign (Attié et al., 2014; Di
Biagio et al., 2015; Sič et al., 2016) held in summer 2012 was dedicated to the export/import of
pollutants from the French continent to the Mediterranean Sea by means of balloon and airborne

measurements. The Aerosol Direct Radiative Impact in the Mediterranean (ADRIMED) campaign investigated aerosols of various origins and their optical properties over the western basin in summer 2013 (Mallet et al., 2016). The Secondary Aerosol Formation in the Mediterranean (SAFMED) campaigns focused on the organic reactive gases and aerosol over the northwestern basin and southeastern France in summer 2013 and 2014 (Di Biagio et al., 2015). Finally, the Gradient in Longitude of Atmospheric constituents above the Mediterranean basin (GLAM) campaign (Ricaud et al., 2017) held in August 2014 was dedicated to the study of the gradient of chemical constituents (pollutants and GHGs) from Toulouse (France) to Larnaca (Cyprus) and the impact of the Asian monsoon anticyclone on the EM pollutant levels.

Surface background stations in the EM (e.g., Crete, Greece and Larnaca, Cyprus) and in the western Mediterranean Basin (e.g., Menorca, Spain and Lampedusa, Italy) deployed even more instruments to obtain a wide variety of atmospheric parameters (meteorology, chemistry, dynamics, radiation, etc.). These campaigns were organized in close relationship with modelling studies (forecasts, and re-analyses) and space-borne observations. New airborne campaigns are under analysis, e.g. Oxydation Mechanism Observation (OMO) in summer 2015, or in project (Radiative Impact of the Arabian Sea pollutants, greenhouse gases and aerosols on the eastern MEditerranean climate in Summer (RIMES) in summer 2019) in order to quantify the export of the Asian pollutants to the EM basin and its impact on the chemical constituents loading.

Concurrently to these intensive experiments, new sites have been instrumented. In early 2015, the Agia Marina Xyliatou EMEP rural background air quality station sited at 532 m in altitude in the center of Cyprus (35.03°N, 33.05°E), and operated since October 1996 (Kleanthous et al., 2014), has been augmented with a package of atmospheric chemistry and physics monitoring instruments thanks to the Cyprus Institute and French laboratories, in order to initiate an enhanced atmospheric chemistry observation period of several years in the easternmost Mediterranean Basin. Unmanned aircraft vehicles are also deployed on a regular basis to document the lower troposphere above the station and the German Leibniz Institute for Tropospheric Research (TROPOS) institute has deployed a full set of aerosol-cloud-water vaper remote sensing instrument for almost a year in October 2016. This unprecedented experimental effort is expected to bring information on the variability of new compounds and processes with a

focus on VOCs and secondary and carbonaceous aerosols and their origins, and on interactions
between aerosols and the water vapor cycle in this region.
Acknowledgments

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

**Table 1.** Monthly long-term means (LTM) and standard deviation (S.D.) of the mixing layer
depth (MLD), wind speed and range of ventilation rates over Beit-Dagan in the central coast of
the EM. LTM and S.D. values for MLD include the years 1955-1968 (Rindsberger, 1974), 1981-
1984 (Dayan et al., 1988), and 1987-1989 (Dayan and Rodniski, 1999). LTM and S.D. values for
wind speeds are from the NCEP/NCAR Reanalysis Project (NOAA- CIRES Climate Diagnostic
Center) for a 51-year data record over 1948-1999 from http://www.esrl.noaa.gov/psd/ (adapted
from Matvev et al., 2002).

| Month | MLD (m) | | Wind Speed (m s$^{-1}$) | | Ventilation Rates (m$^2$ s$^{-1}$) |
|---|---|---|---|---|---|
| | LTM | S.D. | LTM | S.D. | Range of LTM |
| June | 810 | 470 | 5.5 | 2.25 | 1105 – 9920 |
| July | 870 | 450 | 5.0 | 1.65 | 1365 – 8780 |
| August | 820 | 395 | 4.5 | 1.50 | 1275 – 7290 |


**Table 2.** Compilation by Rudich et al. (2008) of sulfate particulate concentrations and yearly
fluxes from [a] Luria et al. (1996), [b] Wanger et al. (2000) and [c] Matvev et al. (2002).

| Regions | Measurement Periods | Conc. Avg (nmole m$^{-3}$) | Yearly Flux (Tg y$^{-1}$)(*) | Authors |
|---|---|---|---|---|
| Judean mountains | July-Aug. 1984,1986 | 86 | 0.08 | [a] |
| | May-June 1989 | 70 | 0.06 | |
| | July-Aug. 1987,1988 | 103 | 0.09 | |
| | July-Aug. 1990 | 128 | 0.12 | |
| | May, July 1990, 1991 | 85 | 0.08 | |
| Sea of Galilee | Aug.-Sept. 1993 | 87 | 0.03 | |
| | Dec. 1993 | 71 | 0.07 | |
| North coastal plain | June 1993 | 106 | 0.12 | |
| Eastern Mediterranean coast | Sept. 1993 | 38 | 0.08 | [b] |
| | June 1994 (**) | 108 | 0.22 | |
| | June 1998 | 105 | 0.16 | |
| | Sept. 1996 | 26 | 0.04 | [c] |
| | Nov. 1995 | 21 | 0.03 | |

(*) following Matvev et al. (2002) conversion from nmole m$^{-3}$ to yearly fluxes takes into account
the vector component of onshore wind speed, length of flight leg, and the MLD.
(**) the June 1994 flight has been performed during a highly-polluted month over Israel.

**Table 3.** Locations and elevations of NOAA Earth System Research Laboratory Global
Monitoring Division (ESRL/GMD) background sites for CO measurements plotted in Figure 19.

| Code | Name | Latitude | Longitude | Elevation (m) | Country |
|------|------|----------|-----------|---------------|---------|
| WIS | WIS Station Negev Desert | 31.13 | 34.88 | 400.0 | Israel |
| HUN | Hegyhatsal | 46.95 | 16.65 | 248.0 | Hungary |
| LMP | Lampedusa | 35.52 | 12.62 | 45.0 | Italy |
| BSC | Black Sea Constanta | 44.17 | 28.68 | 3.0 | Romania |
| OXK | Ochsenkopf | 50.03 | 11.80 | 1022.0 | Germany |
| BAL | Baltic Sea | 55.35 | 17.22 | 3.0 | Poland |
| MHD | Mace Head County Galway | 53.33 | -9.90 | 5.0 | Ireland |


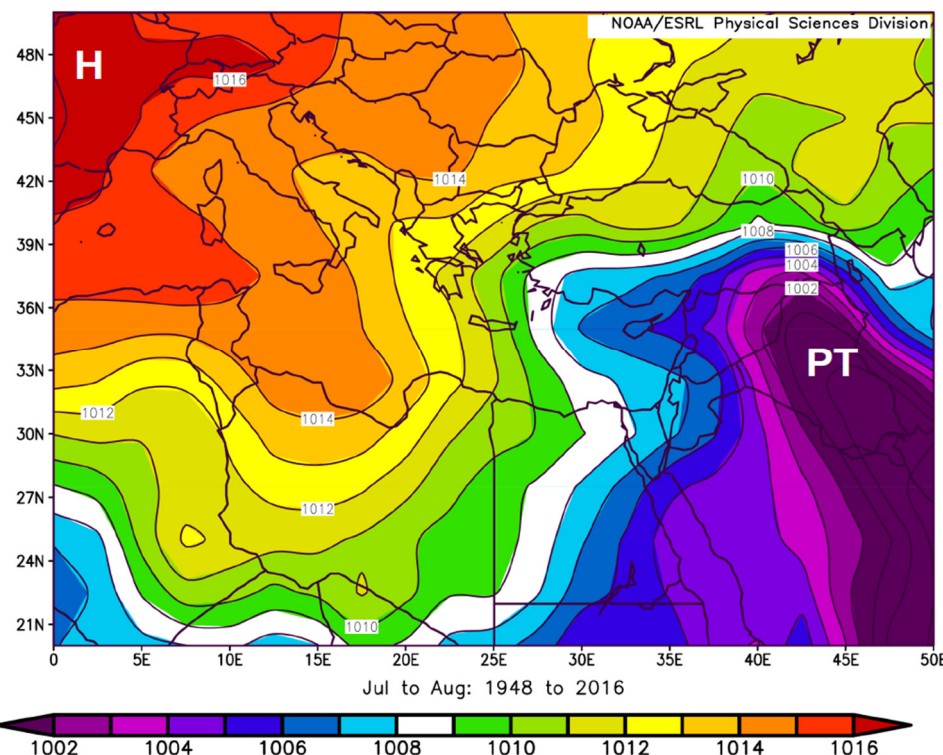

**Figure 1.** Composite long-term mean Sea Level Pressure (hPa) for July-August over 1948-2016. "PT" indicates the Persian Trough position. "H" indicates the Anticyclone position. Source: NCEP reanalysis data provided by the NOAA/OAR/ESRL PSD, Boulder, Colorado, USA, http://www.esrl.noaa.gov/psd/.

1621

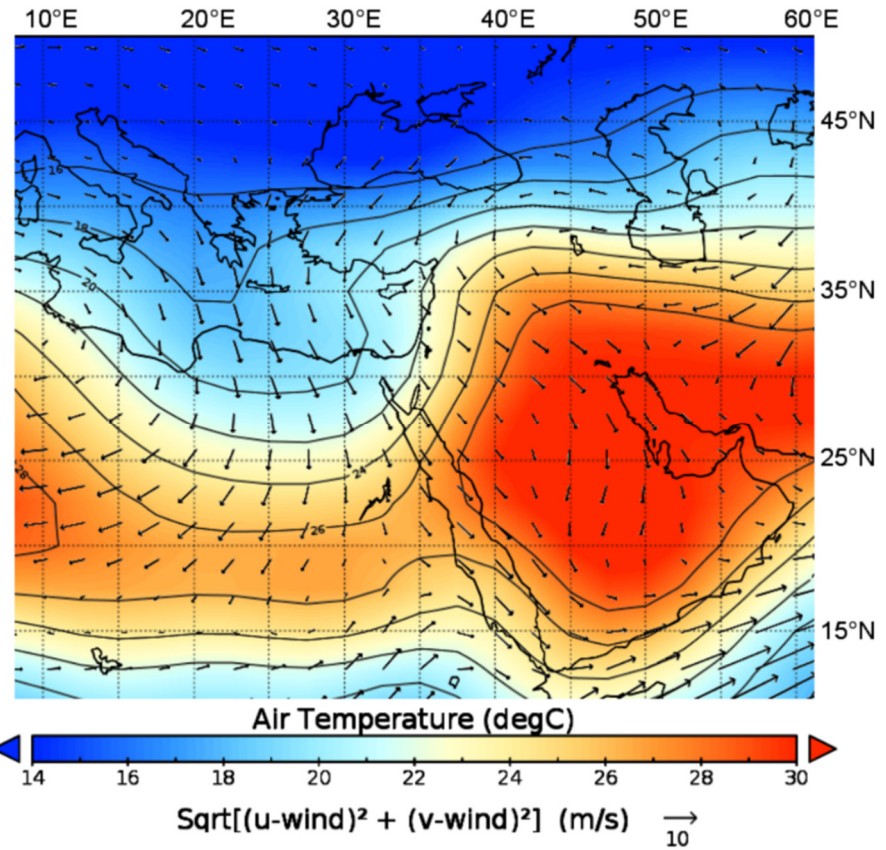

Air Temperature (degC)

Sqrt[(u-wind)² + (v-wind)²] (m/s)

1622

**Figure 2.** NCEP/NCAR reanalysis composite long-term mean temperature at 850 hPa (~1500 m, above sea level or a.s.l.) with wind vectors, averaged over 1948-2016 for July-August. Note the southward penetration of the Europe an cold air over the Mediterranean Basin. This cold air mass is transported at shallow tropospheric layers towards the Eastern Mediterranean by the Etesian northwesterlies characterizing the Persian trough. Source: NCEP reanalysis data provided by the NOAA/OAR/ESRL PSD, Boulder, Colorado, USA, http://www.esrl.noaa.gov/psd/.


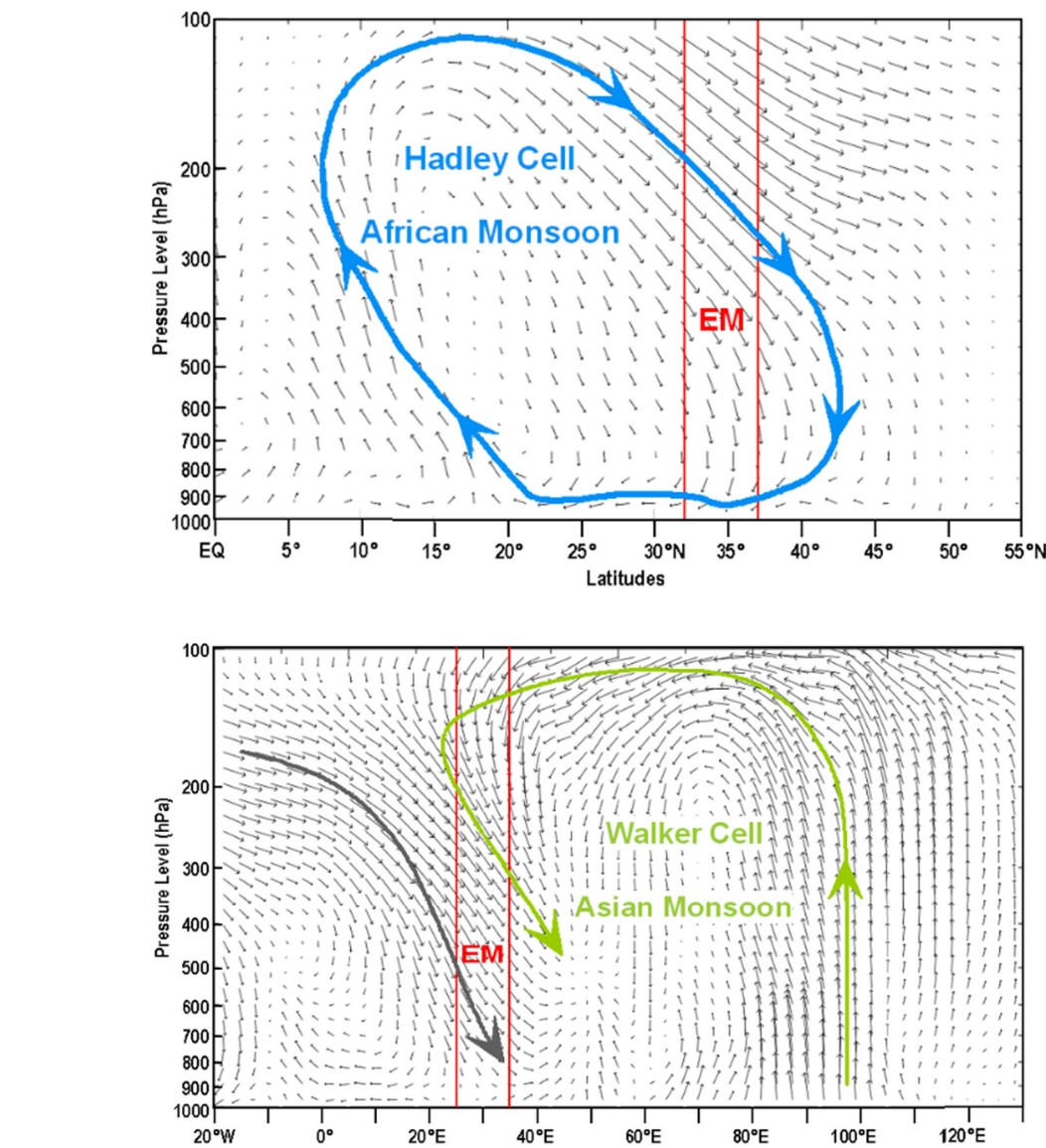



**Figure 3.** (Top) Closed Hadley cell circulation of the African monsoon depicted by the vertical cross section of wind vectors for July-August averaged over the 30-40°E longitudinal band. (Bottom) Closed Walker cell circulation of the Asian monsoon depicted by the vertical cross section of wind vectors for July-August averaged over the 20-35°N latitudinal band. The two figures are based on the NCEP/NCAR long-term averages (1948-2016) with the position of the eastern Mediterranean (EM) in red. Source: NCEP reanalysis data provided by the NOAA/OAR/ESRL PSD, Boulder, Colorado, USA, http://www.esrl.noaa.gov/psd/.









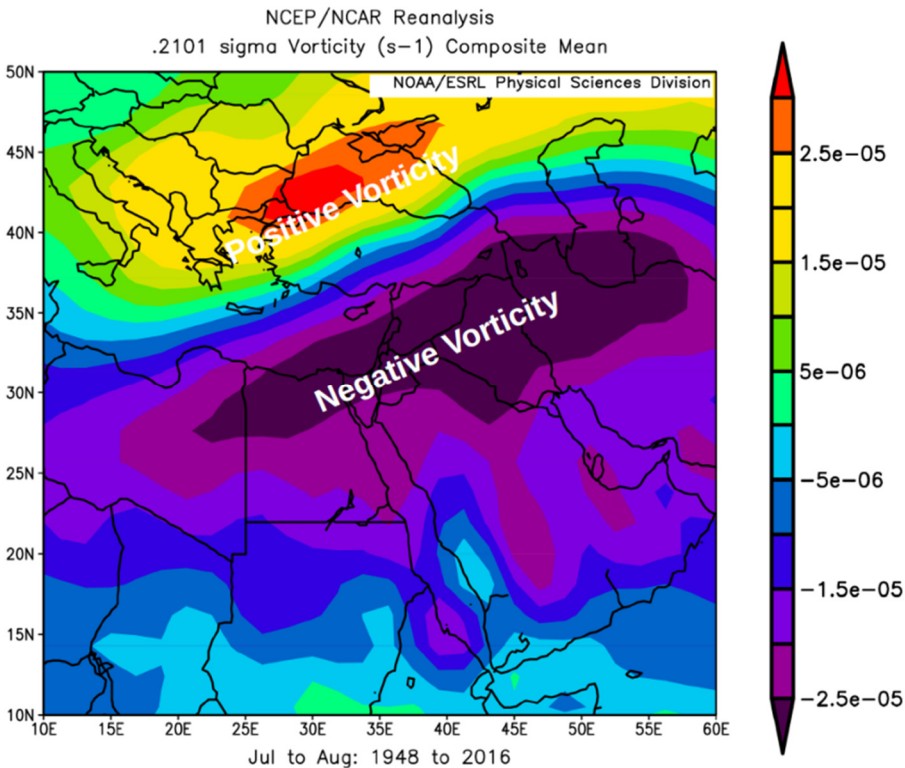


**Figure 4.** NCEP/NCAR Reanalysis long-term averages (1948-2016) of the relative vorticity at 200 hPa (~12 km a.s.l.) for July-August. The relative vorticity vector is generally perpendicular to the ground, positive when the vector points upward, negative when it points downward. Note the negative relative vorticity region located over the southeastern Mediterranean as a result from both shear and curvature negative relative vorticity. Relative vorticity units are $10^{-5}$ $s^{-1}$. Source: NCEP reanalysis data provided by the NOAA/OAR/ESRL PSD, Boulder, Colorado, USA, http://www.esrl.noaa.gov/psd/.


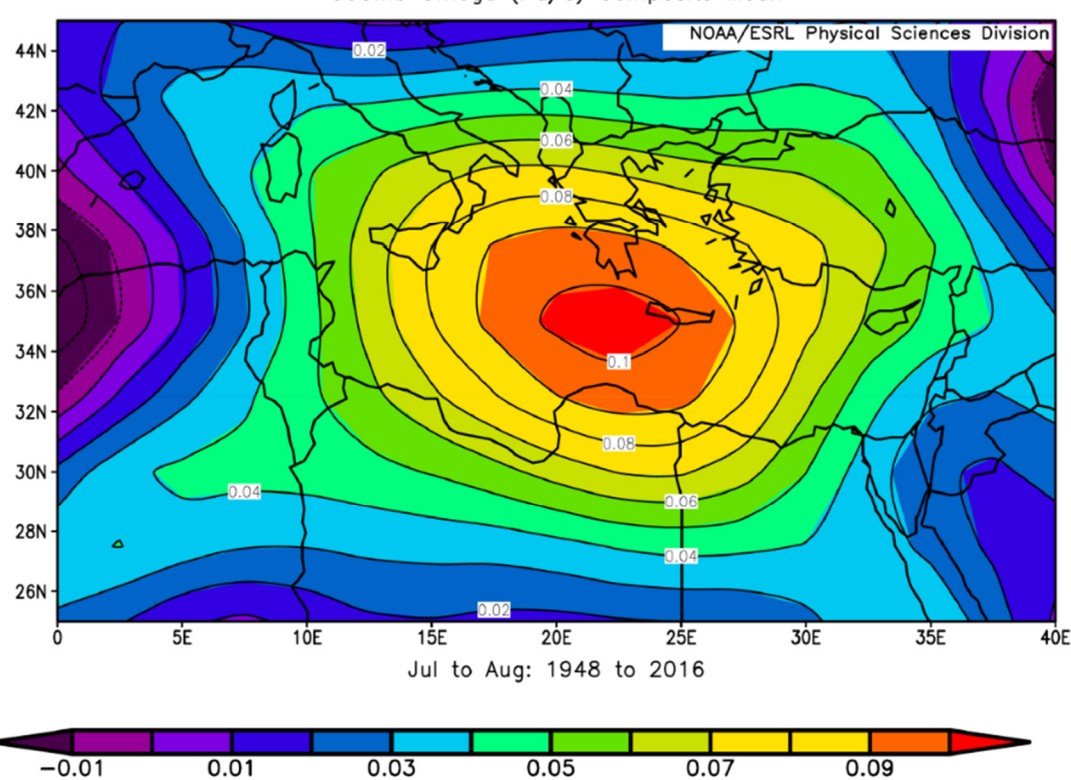


**Figure 5.** NCEP/NCAR reanalysis long-term averages of Omega (Pa s$^{-1}$) at 500 hPa (~5.5 km a.s.l.) designating vertical motion for July to August 1948-2016. The maximum subsidence of 0.1 Pa s$^{-1}$ is equivalent to a downward air motion of ~1.5 cm s$^{-1}$. Source: NCEP reanalysis data provided by the NOAA/OAR/ESRL PSD, Boulder, Colorado, USA, http://www.esrl.noaa.gov/psd/.



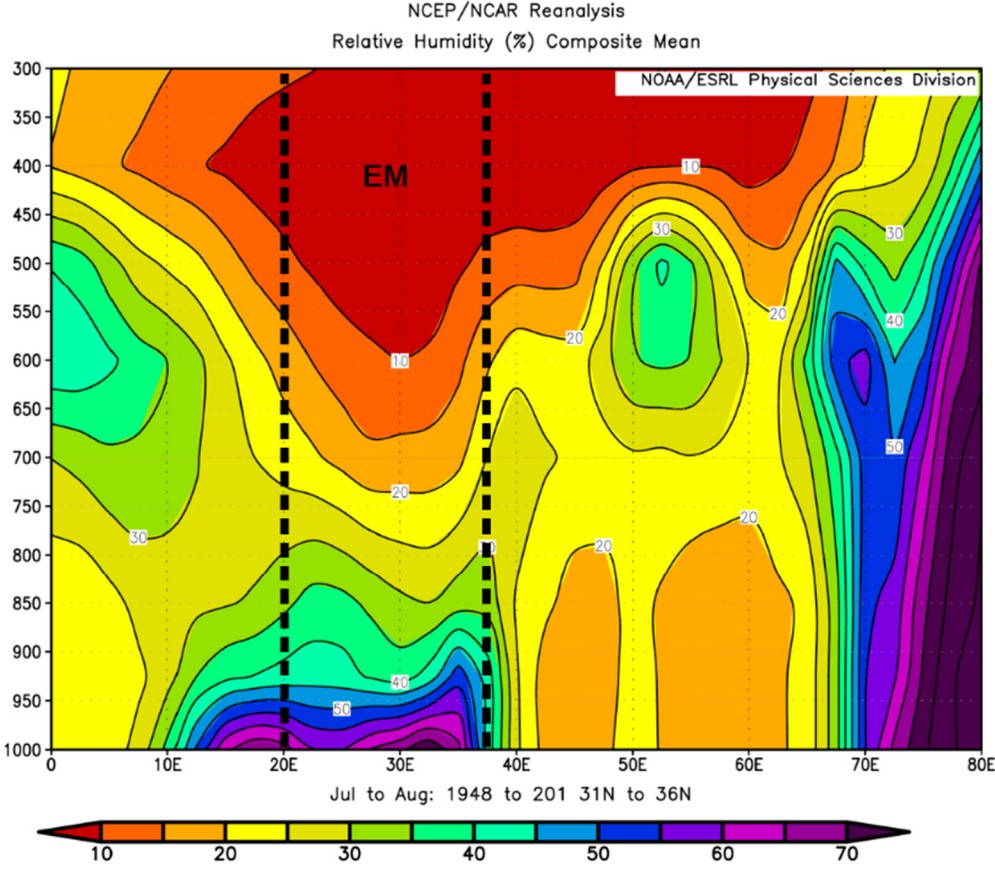

Figure 6. Long-term mean vertical cross section of relative humidity, averaged over the 31-36°
N latitudinal band for July-August 1948-2016 with the eastern Mediterranean position (EM), in
dashed black lines. Source: NCEP reanalysis data provided by the NOAA/OAR/ESRL PSD,
Boulder, Colorado, USA, http://www.esrl.noaa.gov/psd/.

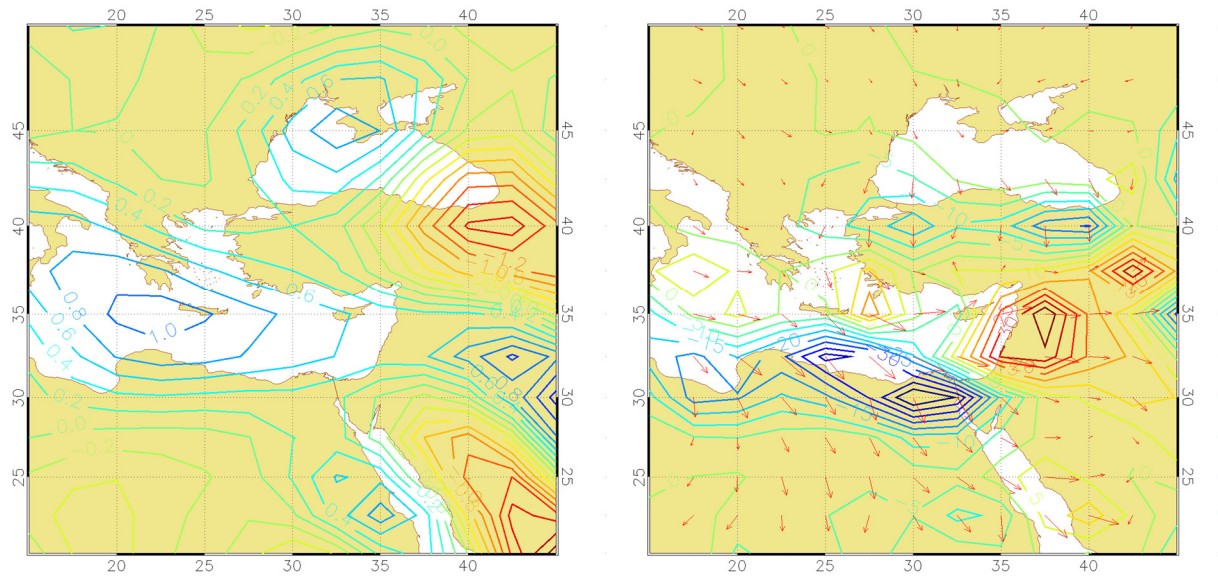

1662

**Figure 7.** (Left) Blue contours display positive Omega values (cm s$^{-1}$) representing the vertical descending air motion at a mid-tropospheric level (700 hPa) (~3 km a.s.l.) pointing at a core of 1 cm s$^{-1}$ located over Crete. Red contours are negative Omega values. (Right) Blue contours display cold advection calculated as multiplication of the horizontal thermal gradient by the wind vector. Red contours indicate warm advection, both at 995 hPa level, equivalent to about 140 m a.s.l at 12:00 UTC during Persian trough summer synoptic conditions. Source: NCEP reanalysis data for 2000-2012, provided by the NOAA/OAR/ESRL PSD, Boulder, Colorado, USA, http://www.esrl.noaa.gov/psd/.


**Asian Monsoon Intensification**

Upward air motion increases over Asia → Subsidence increases over EM → Adiabatic heating over EM

Surface pressure decreases over Asia → Etesian wind strengthening → Cold advection increases over EM



**Figure 8.** Schematic of the proposed mechanism during intensification of the Asian monsoon
(reproduced from Ziv et al., 2004).

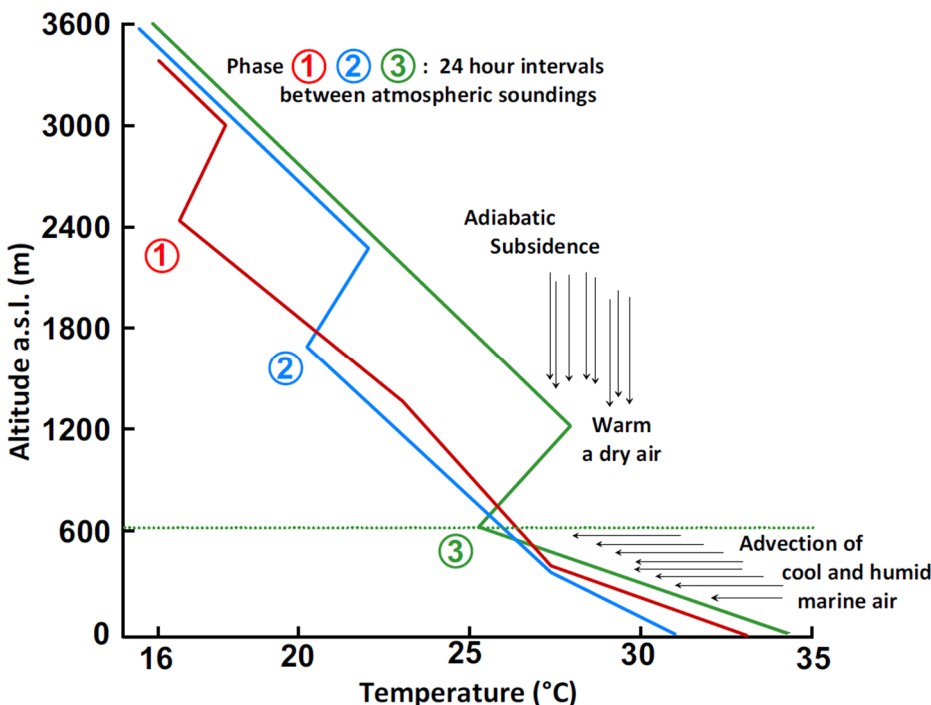


**Figure 9.** Successive schematic sounding thermal profiles indicating the downward motion of
adiabatic subsidence accompanied by a weakening of the Persian Trough, which restricts the
mixing layer depth to shallow layer of the atmosphere (phases 1-3 are 24 h intervals between
each sounding at Beit-Dagan, Israel); (from Dayan et al. (1988); ©American Meteorological
Society; used with permission).


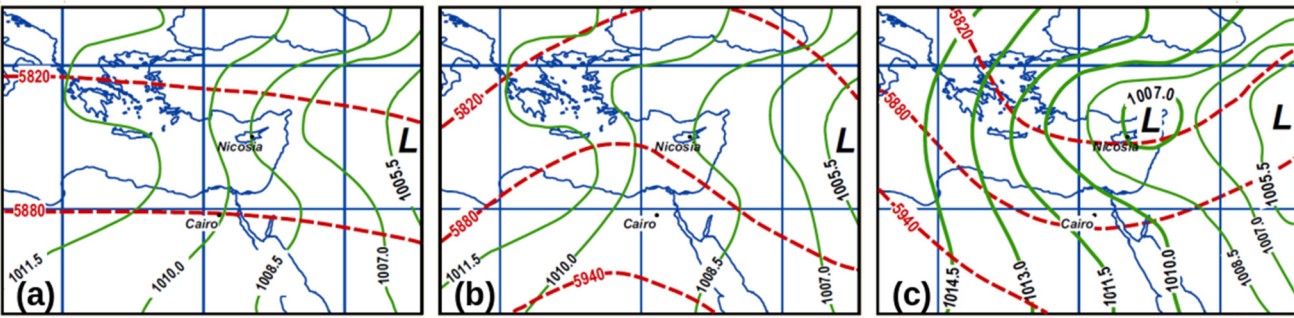

**Figure 10.** Typical synoptic charts showing the three modes: (a) moderate, (b) shallow, and (c)
deep mode of the Persian Trough as defined by the surface-pressure differences between Nicosia
(Cyprus) and Cairo (Egypt), and their associated upper level conditions. Solid lines are isobars of
sea level pressure with 1.5-hPa intervals. Dashed lines are contours at 500-hPa level with 60-m
intervals (from Dayan et al. (2002); ©American Meteorological Society; used with permission).

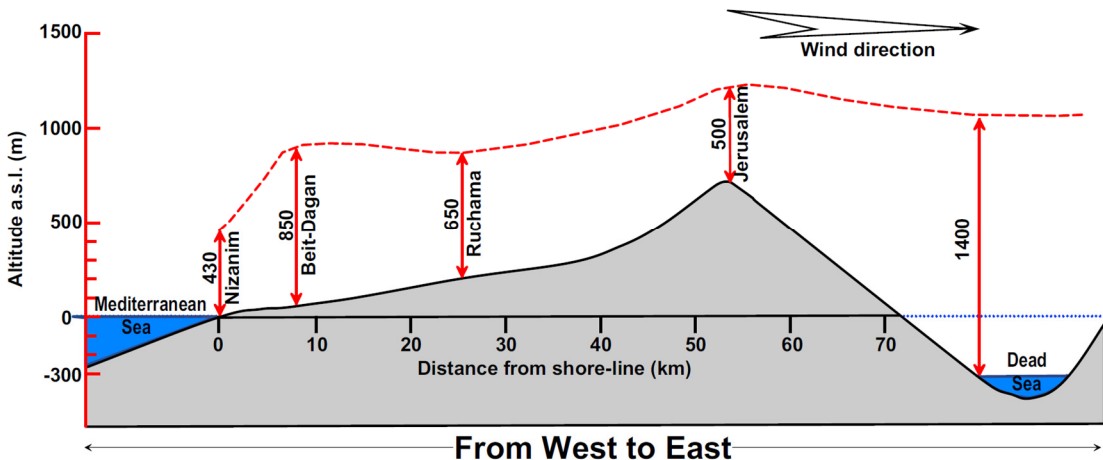


**Figure 11.** Schematic description of the lateral variation of the mixing layer depth (m, a.s.l) from
the Mediterranean Sea to the Dead Sea (from Dayan et al. (1988); ©American Meteorological
Society; used with permission).

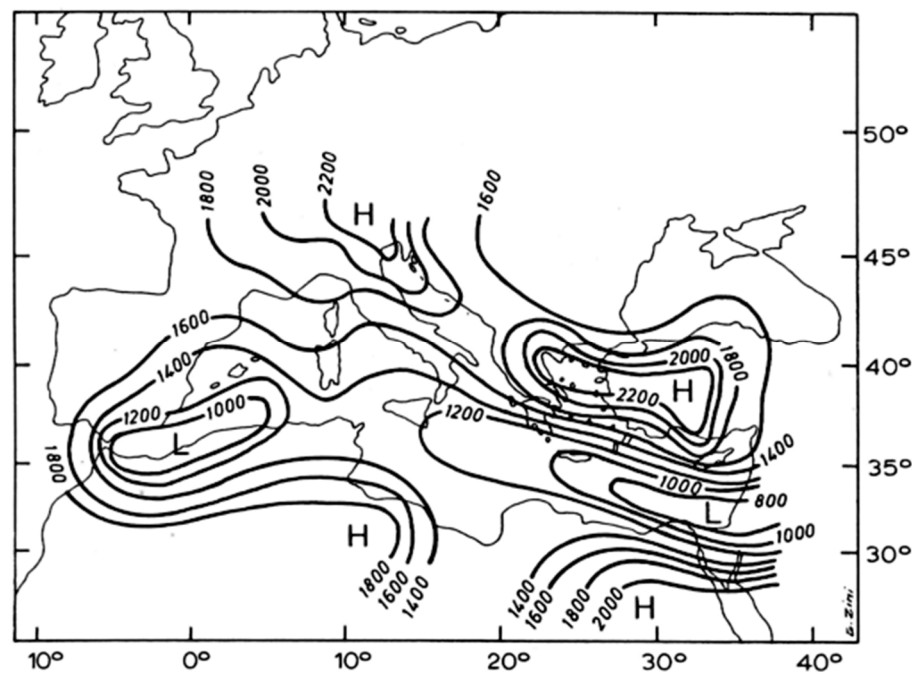


**Figure 12.** Seasonal map of the mixing layer depth (m) for summer (June, July, August) 1987
over the Mediterranean region at 12:00 UTC (from Dayan et al. (1996), permission requested
from Kluwer Academic Publishers).

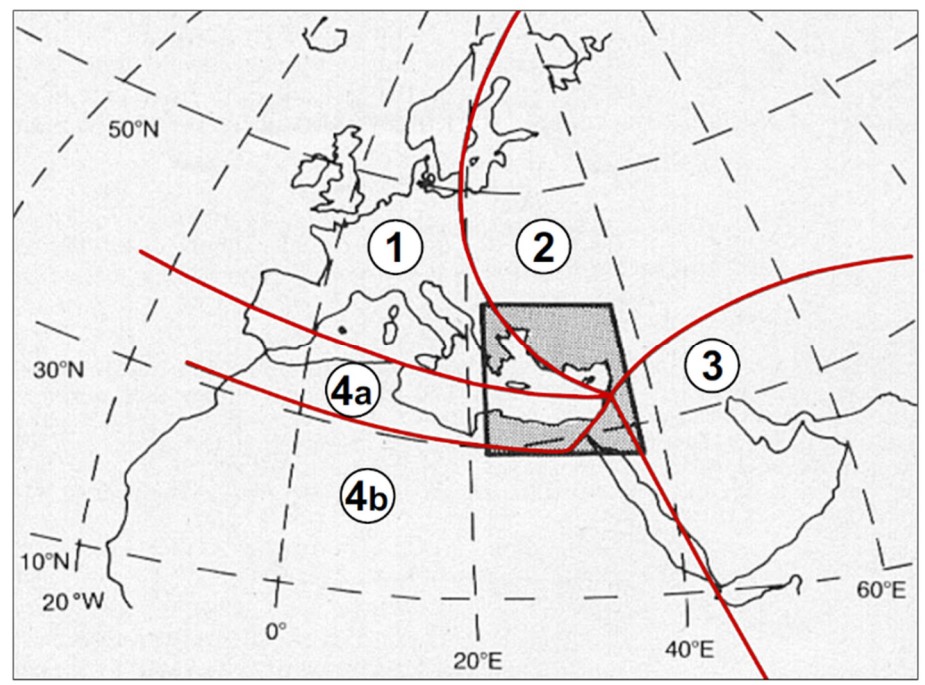


**Figure 13.** Trajectory typing method used to categorize 5-day back-trajectories from the Eastern Mediterranean region at 850 hPa using the Air Resources Laboratory's trajectory model GAMBIT over the 1978-1982 period (from Dayan (1986), ©American Meteorological Society; used with permission).



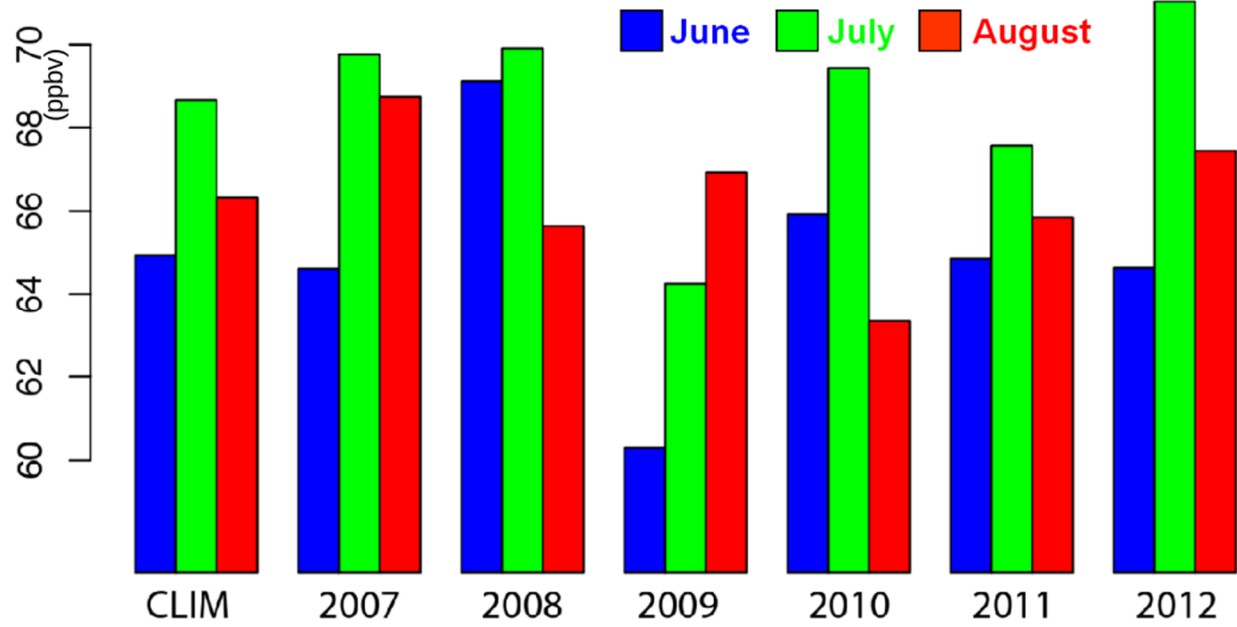


Figure 14. June, July and August monthly means of $O_3$ concentrations (ppbv) at 3 km partial column measured by IASI in summer (June, July, August) within the 2007-2012 period over the Mediterranean (IASI morning overpasses). Only the observations over the sea are considered in the averages. The monthly means referred to as "CLIM" represent the averages over the whole period (adapted from Doche et al., 2014).

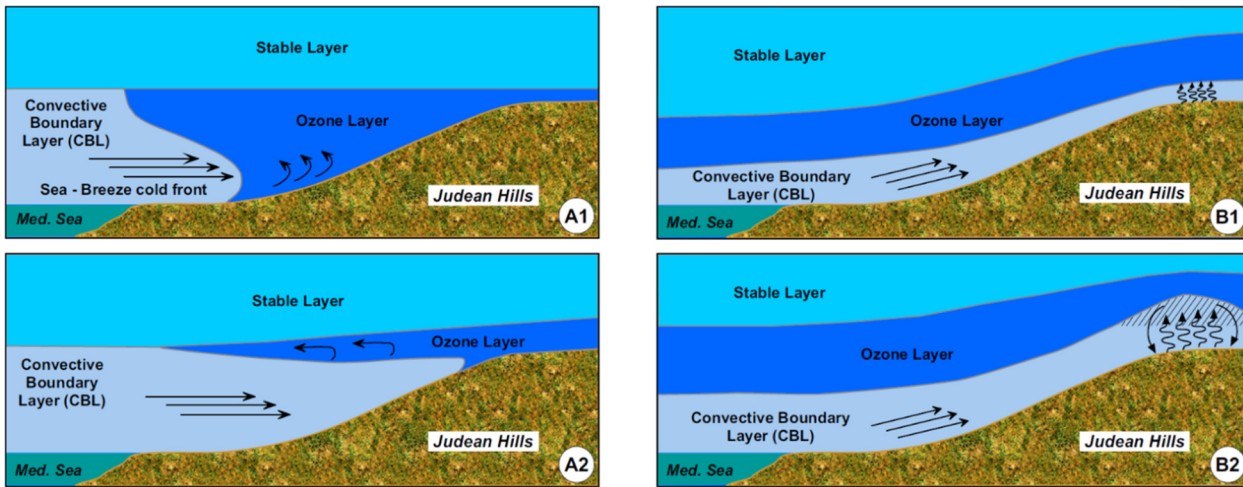

**Figure 15.** Scheme of the mechanism causing fumigation of a rich $O_3$ cloud toward the ground as moving inland over the Eastern Mediterranean coast during the weakening of a deep mode of the Persian Trough (from Dayan and Koch (1996); ©American Meteorological Society; used with permission).

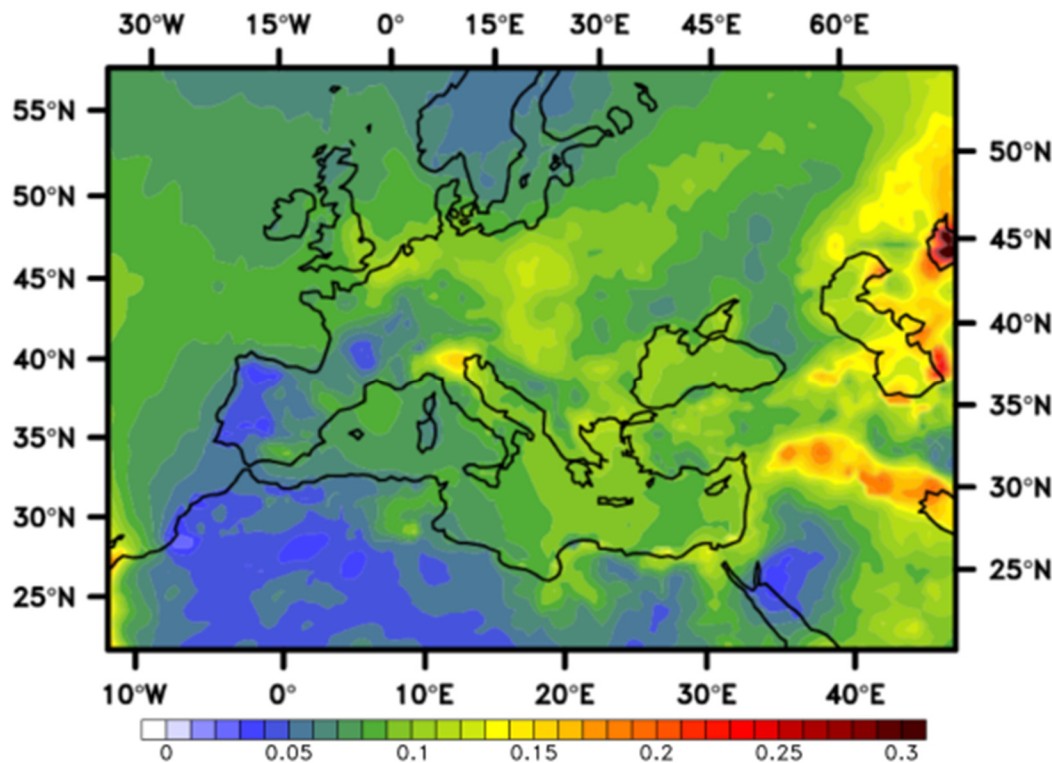


**Figure 16:** Average aerosol optical depth (AOD) contributed by particulate sulfate validated
against AERONET AOD observations over the period 2003–2009. As mentioned by
http://www.esrl.noaa.gov/gmd/grad/surfrad/aod/, a value of 0.01 corresponds to an extremely
clean atmosphere, and a value of 0.4 to a very hazy condition (the 2003-2010 average AOD over
the Mediterranean Basin is ~0.20) (adapted from Nabat et al., 2013).

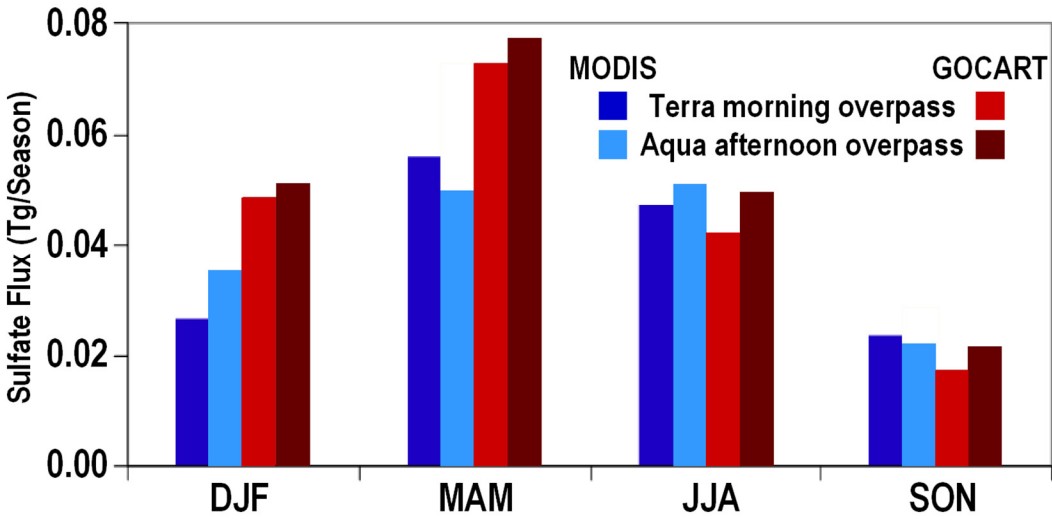


**Figure 17.** Seasonal flux (Tg season‑1) of dry sulfate as derived from MODIS/Terra and MODIS/Aqua space-borne observations compared to GOCART model-derived results, along the 150-km Israeli coastline of the Eastern Mediterranean Sea. Seasons on Xaxis are: winter (DJF), spring (MAM), summer (JJA), autumn (SON). (adapted from Rudich et al., 2008).

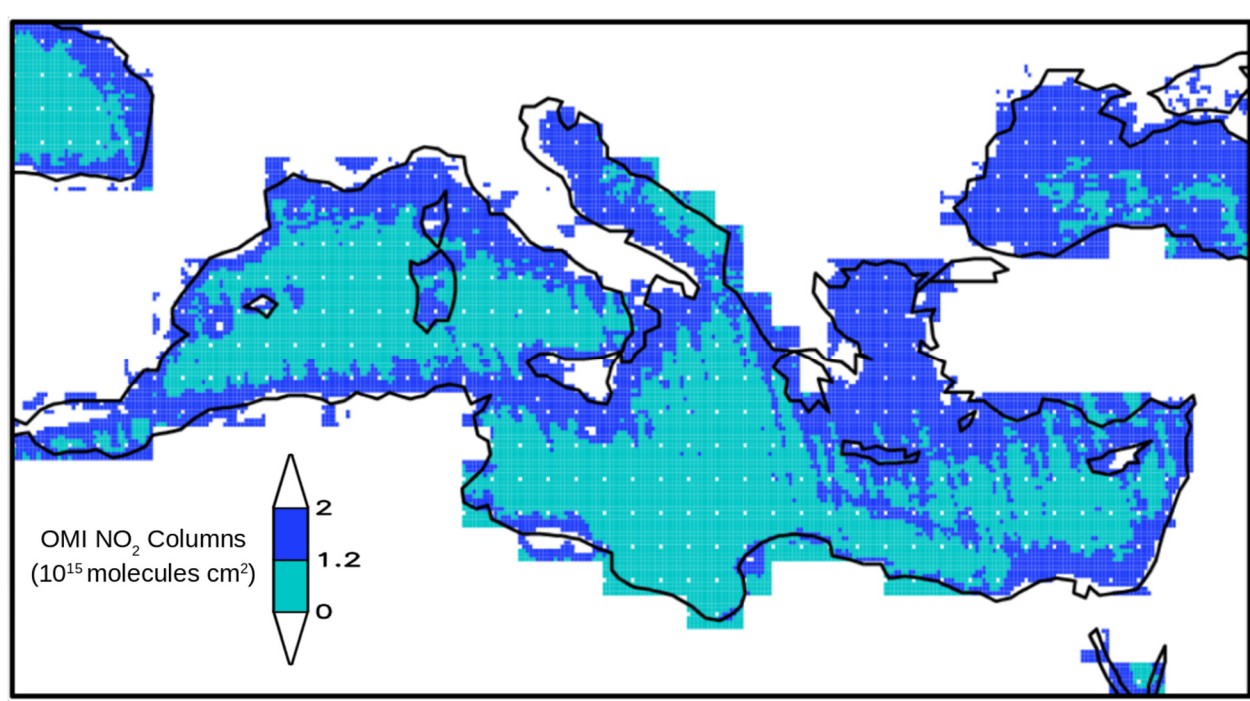

1735

**Figure 18.** Seasonal average over June-August 2006 of OMI NO₂ columns over the Mediterranean Sea ($10^{15}$ molecules cm$^{-2}$), retrieved from the OMI satellite and considering only maritime pixels (reproduced from Marmer et al., 2009).

1739

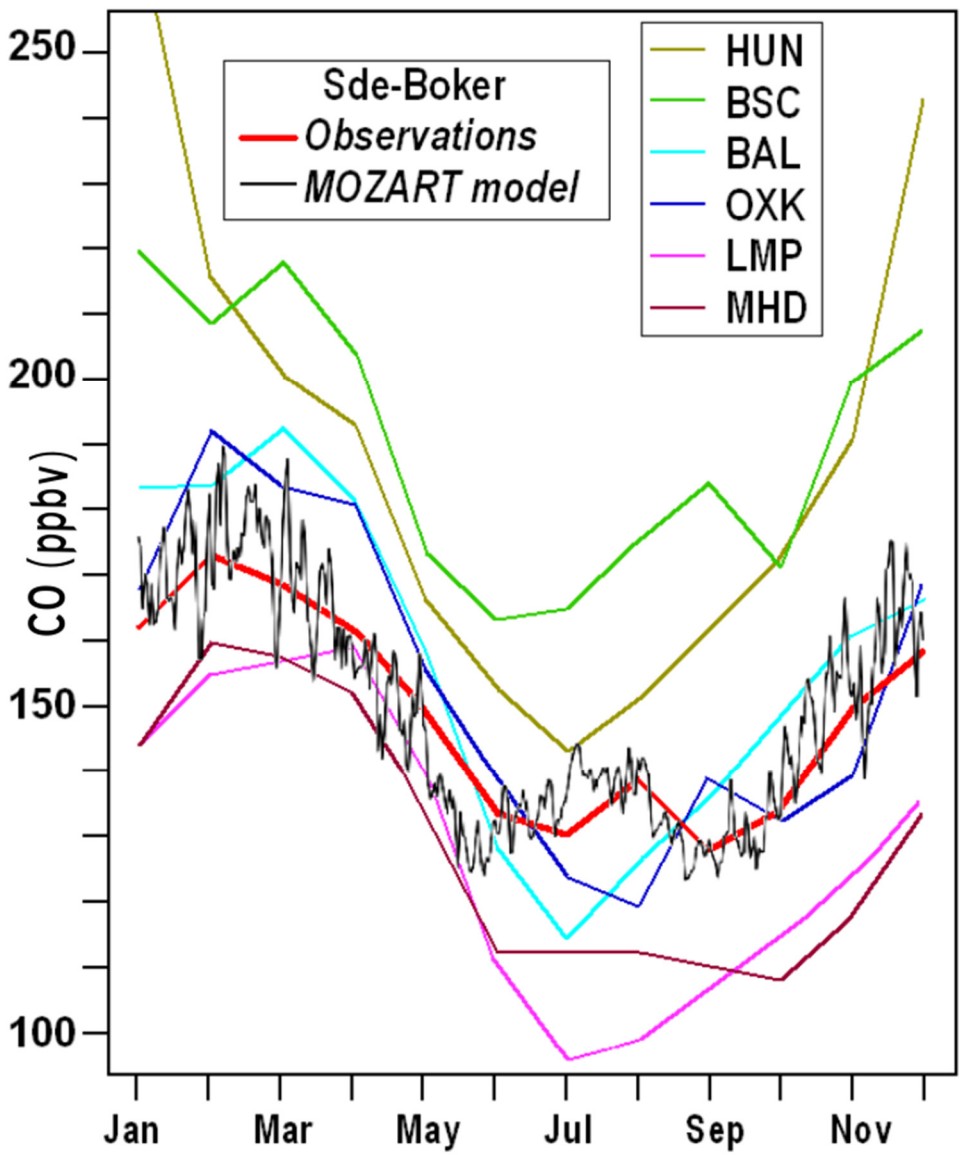

1740

Figure 19. Monthly mean CO concentrations over 1996-2009 at Sde-Boker (red) and at seven
European ESRL/GMD background stations (listed in Table 3, multiple colors), compared to the
five-year averaged CO surface concentrations at Sde-Boker (black) over 2003-2007 from the
MOZART-4 chemistry-transport model (adapted from Drori et al., 2012).






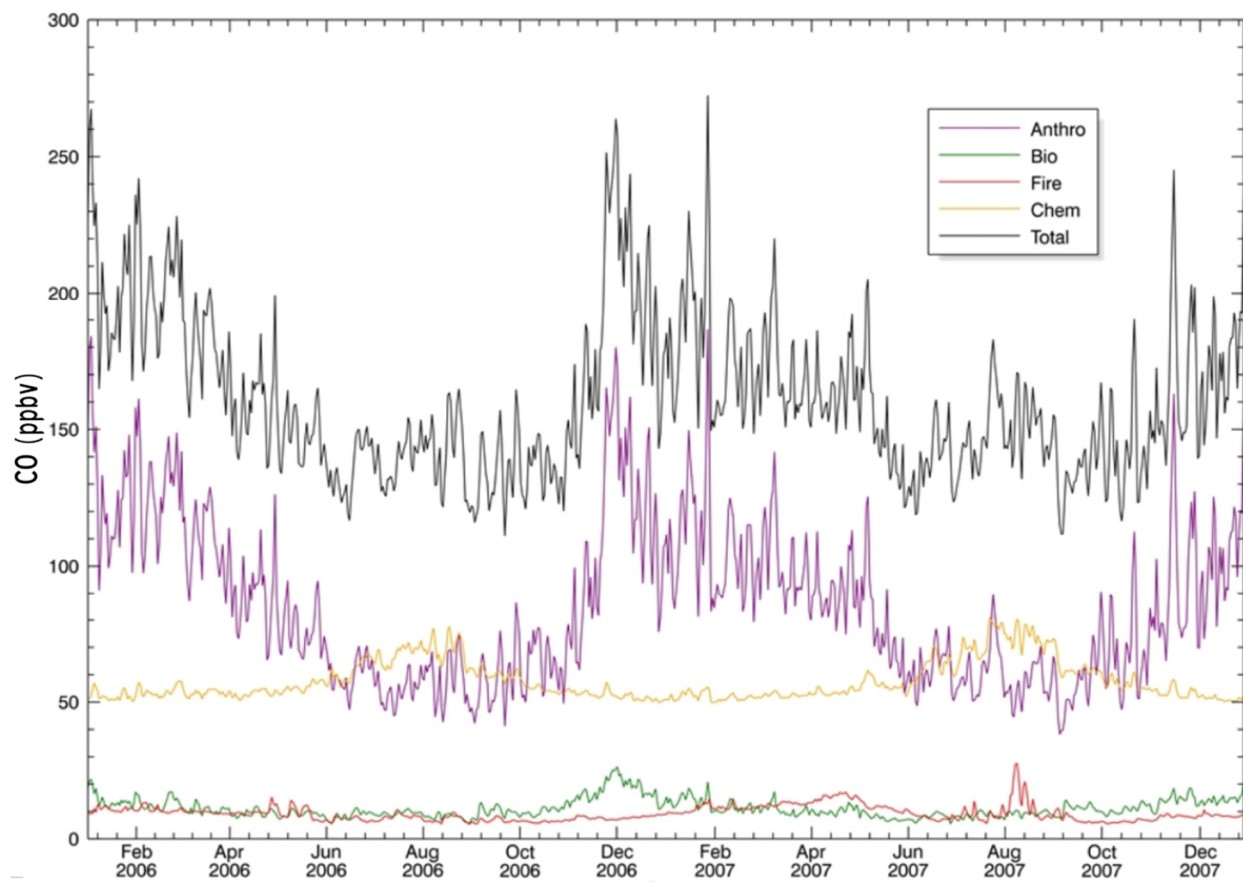

**Figure 20.** Monthly timeseries of total surface CO (black) at Sde-Boker, Israel, and contributions
from specific sources (anthropogenic in purple, chemical production in orange, biogenic in
green, and fires in red; ocean is negligible and not shown) as simulated by MOZART for 2006–
2007 (adapted from Drori et al., 2012).






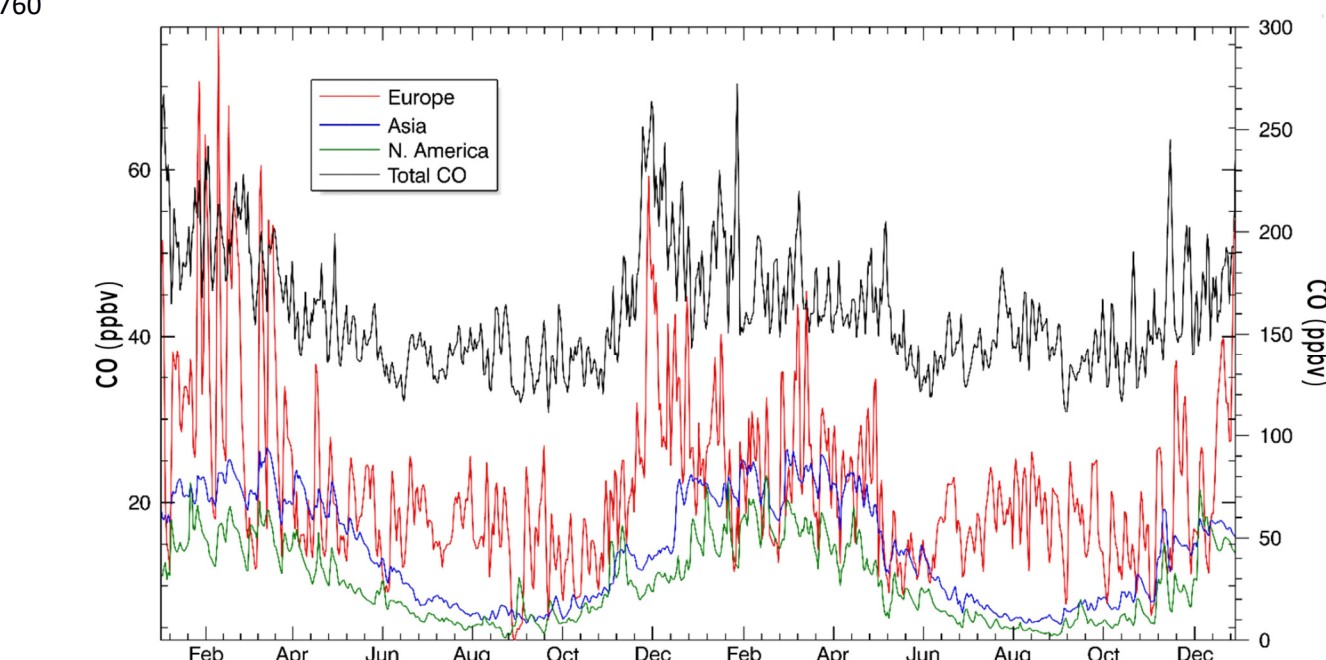


**Figure 21.** Monthly timeseries of the European (red), Asian (blue) and North American (green)
anthropogenic contribution to the total surface CO (black) at Sde-Boker as simulated by
MOZART over 2006-2007. Distinct continents are scaled on the left vertical axis and total CO
on the right vertical axis (adapted from Drori et al., 2012).




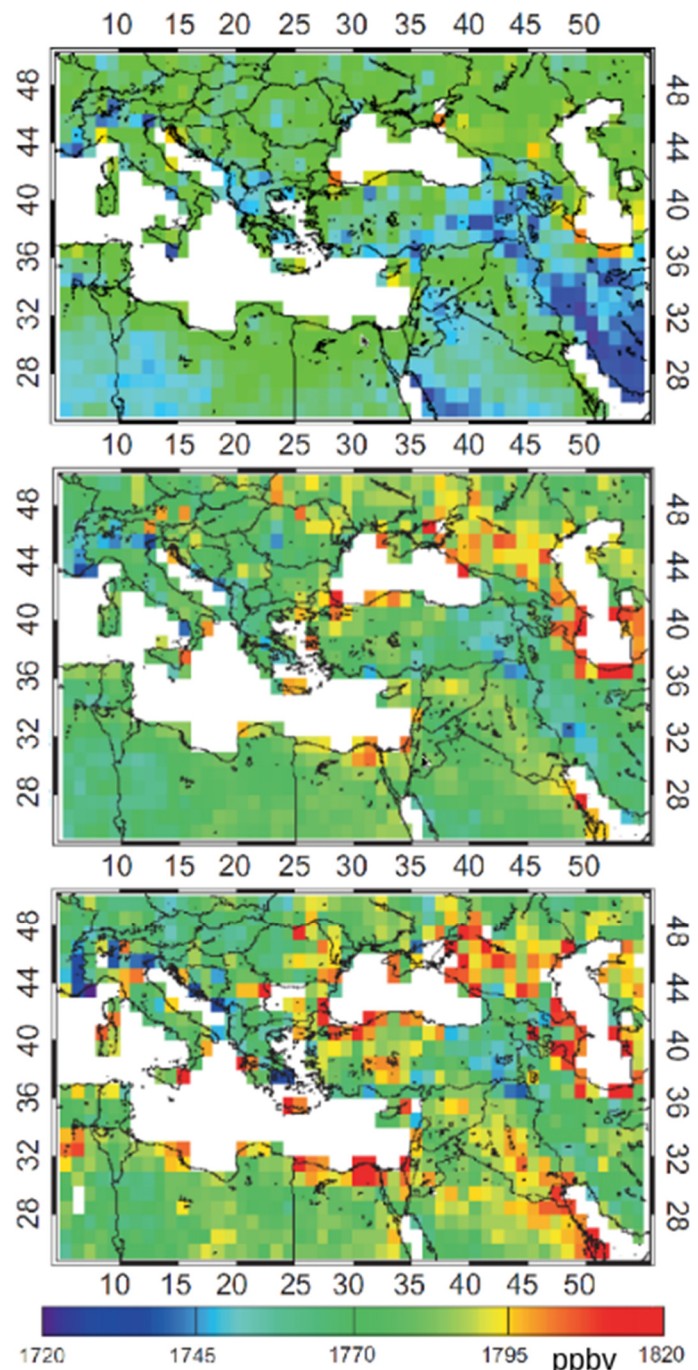


**Figure 22.** Maps by $1° \times 1°$ resolution of dry air column-averaged mole fractions, denoted as SCIAMACHY WFM-DOAS $XCH_4$ levels in 2003 including a yearly average (top panel), a summer average (mid-panel) and an August average (bottom panel), in ppbv (from Georgoulias et al. (2011), permission requested from Taylor and Francis).



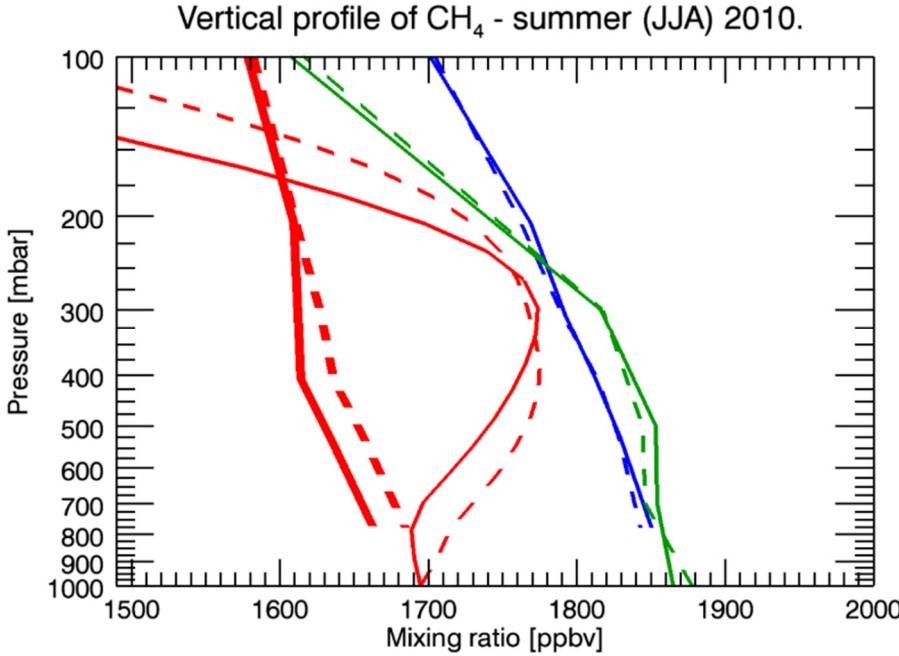


**Figure 23.** Summer averaged vertical profiles of CH$_4$ as measured by AIRS (blue lines) and GOSAT (green lines), and as calculated by MOCAGE (thin red lines) over the eastern (dashed lines) and western (solid lines) Mediterranean Basins in summer 2010. Also shown are the seasonally-averaged MOCAGE profiles convolved with the AIRS averaging kernels (thick red lines) for the summer over the eastern (dashed lines) and western (solid lines) Mediterranean Basins (adapted from Ricaud et al., 2014).



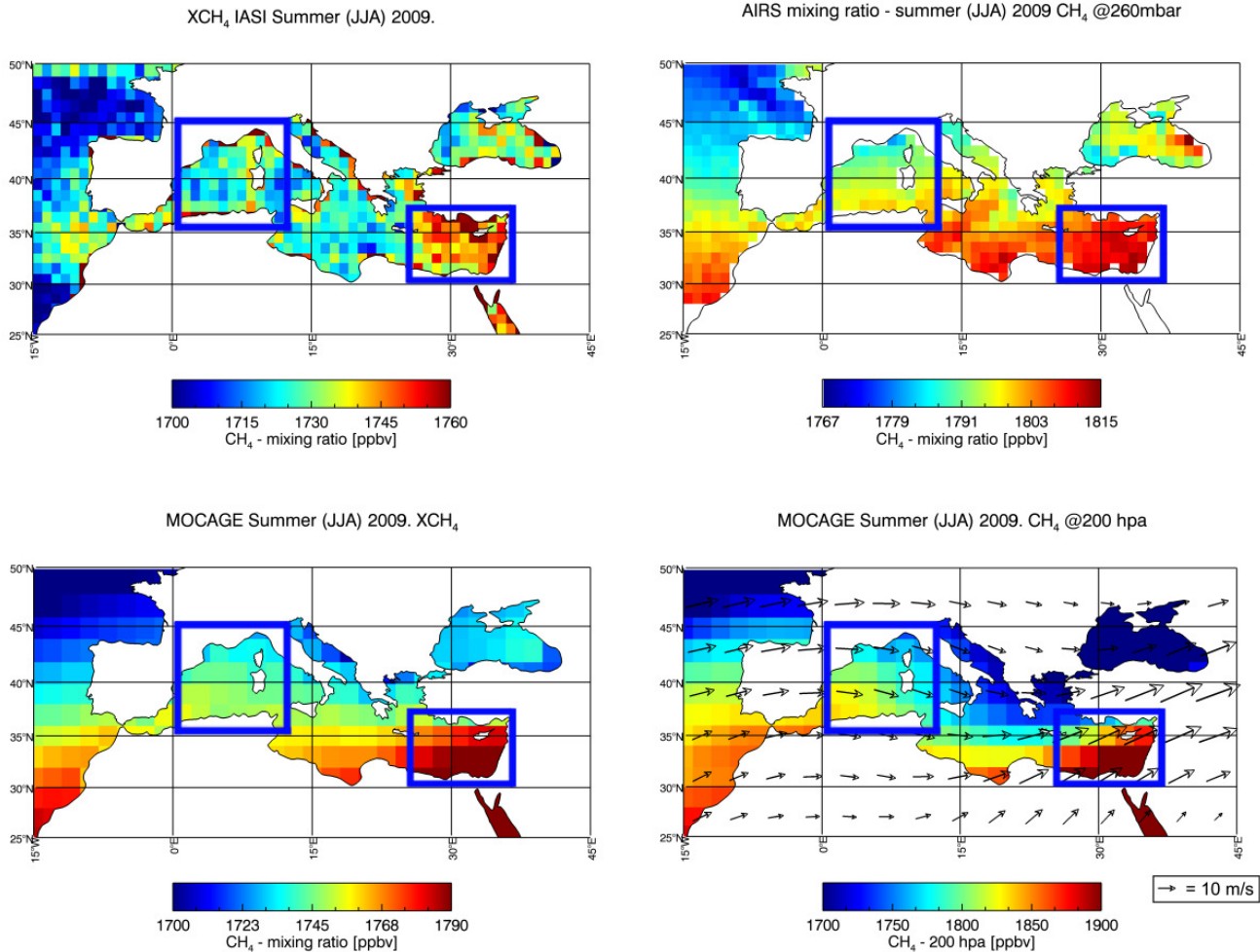


**Figure 24.** Fields of CH₄ as calculated by MOCAGE (bottom) and as measured by IASI (top
left) in total column and AIRS (top right) at 260 hPa averaged for summer (July, July, August)
2009. Horizontal winds are from ARPEGE averaged over the same period. The two blue squares
represent the West and East Mediterranean Basins (adapted from Ricaud et al., 2014).


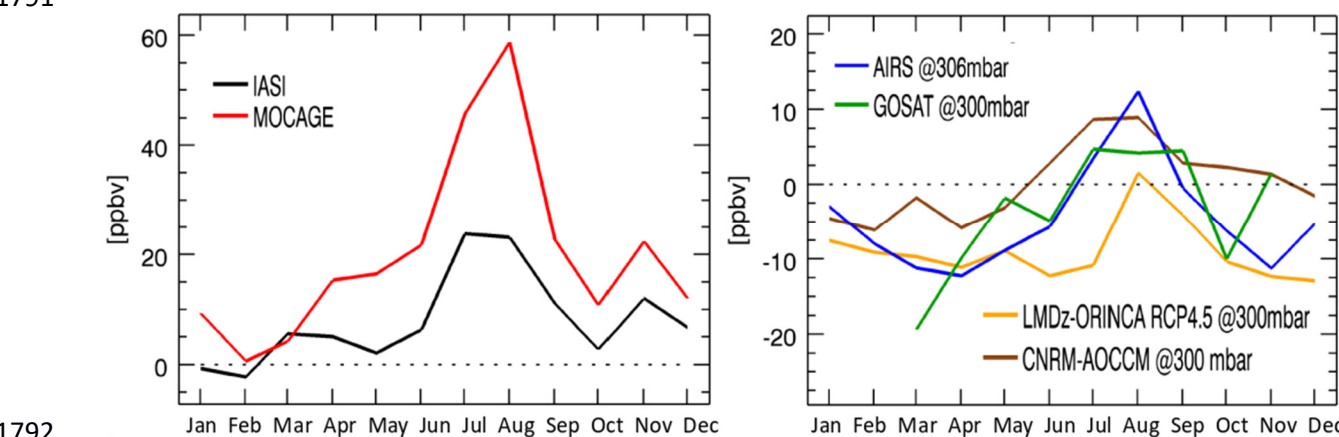


Figure 25. Seasonal evolution of the difference in CH<sub>4</sub> fields between the eastern and western Mediterranean Basin: (right) around 300 hPa as measured by AIRS (blue) and GOSAT (green) and as calculated by LMDz-OR-INCA (yellow) and CNRM-AOCCM (brown), and (left) in total column as measured by IASI and calculated by MOCAGE (adapted from Ricaud et al. 2014).


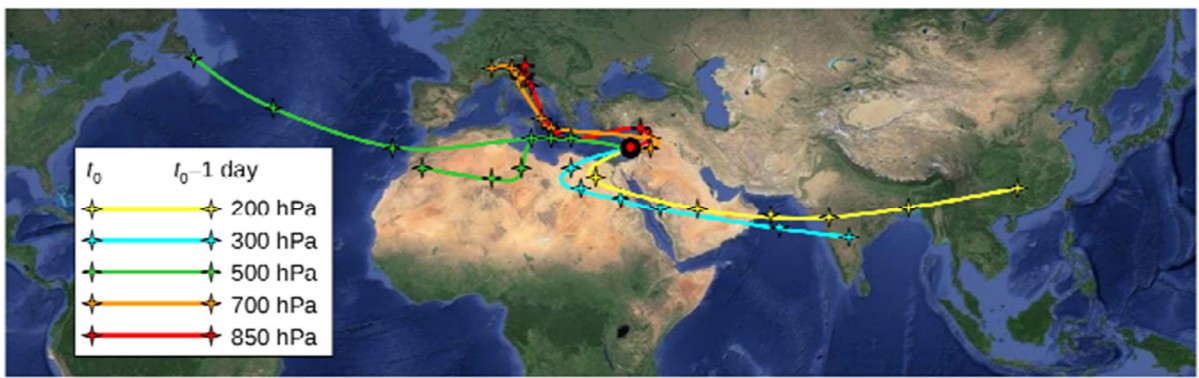


**Figure 26.** Six-day back-trajectories climatology from the point at 33°N and 35°E located off
Israel in the eastern Mediterranean Basin (red filled circle) derived for July-August over 2001-
2010 every 12 hours. The position of the gravity center of each distribution (i.e. the maximum in
the probability density function) at each level is represented every 24 h by a star (adapted from
Ricaud et al., 2014).

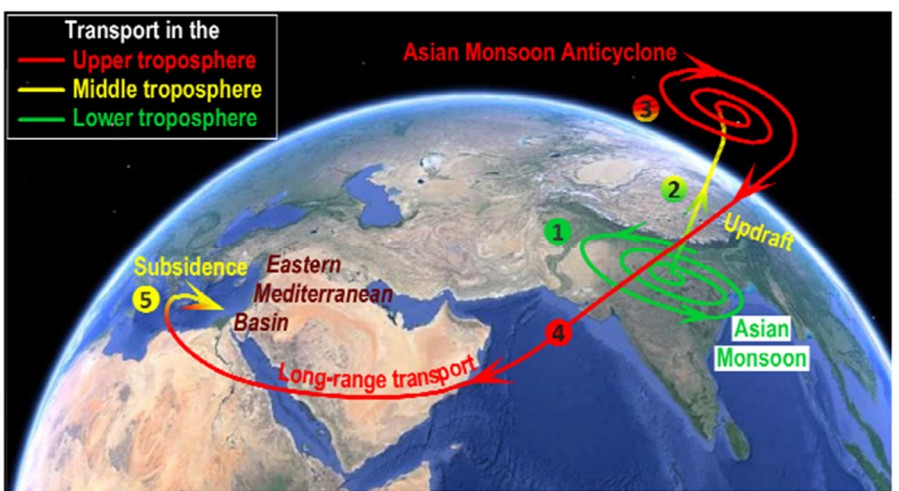


Figure 27. Schematic representation of the processes impacting the mid-to-upper tropospheric pollutants, including $CH_4$ above the Eastern Mediterranean in summer (July-August) (adapted from Ricaud et al., 2014).
