# Peer review of "Atmospheric pollution over the eastern Mediterranean during summer – A review"

_Atmospheric Chemistry and Physics, 2017_

## Referee Comment (RC1) · Anonymous Referee #3 · 28 Apr 2017

This manuscript intends to present a review of atmospheric pollution over the Eastern Mediterranean during summer. There is a lot of information provided from others work but often this information is fragmented, not well connected and generally the manuscript lacks of coherence. Furthermore there is strong self-citation and the vast majority of the discussed articles refer to a specific region of eastern Mediterranean thus leading to an unbalanced discussion especially in some sections. There is no distinction in the discussion of the short-lived species and the long-lived species. Generally, I think that the manuscript needs to be restructured in order to provide all this interesting information in a more coherent way. Please find below a number of major comments that have to be taken into consideration for restructuring the manuscript.

Comments 1) Section 2.1, page 4, line 15: The authors state that dry north Etesian winds are generated by the Persian Trough. It is actually generated by the east–west

pressure gradient manifested by large scale circulation features, low pressures over eastern Mediterranean/Middle East as an extension of the PT and the high pressure over central and southeastern Europe .

2) Section 2.1, page 4, lines 16-21: The discussion for the eastern Mediterranean subsidence during summertime needs elaboration in connection to the discussion in page 5 (lines 16-19) based on the paper of Rodwell and Hoskins. The current consensus view recognises the importance of the interaction with the mid-latitude westerlies of an equatorially trapped Rossby wave to its west induced by the South Asian monsoon heating as well as an enhancement of the descent due to diabatic radiative cooling under clear sky conditions (Rodwell and Hoskins 1996, 2001; Tyrlis et al. 2012).

3) Section 2.1, page 4, line 25: The discussion for the eastward progression of the subtropical high needs clarification. Which subtropical high do the authors mean? During summer the Azores High moves westward toward Bermuda (when it is known as the Bermuda high). Furthermore a number of studies point out the differences between the acticylonic center over central and southeastern Europe causing the Etesians and the Azores permanent Anticyclone (Prezerakos, 1984; Tyrlis and Lelieveld, 2013; Anagnostopoulou et al., 2014). The acticylonic center over central and southeastern Europe causing the Etesians is related primarily with anticyclonic vorticity advection from Northwestern Africa and secondly with diabatic cooling under clear skies.

4) Section 2.2, page 6, line 11: The authors state " Since this turbulent layer is mainly governed by synoptic-scale circulation patterns ..." . What exactly do the authors mean? Please clarify. Is this a general comment or a comment associated with the specific cited study of Dayan et al., 1988?

5) Section 2.2: There is extensive description of the link between synoptic patterns and the structure of the mixing layer in Israel within this session. As a reader I am rather confused and I do not really see the scope of such extended description of this link for a specific region in the frame of an overview paper for the regional baseline atmospheric

pollution concentrations over Eastern Mediterranean.

6) Sections 2.2: The majority of the discussed articles refer to studies at the coast of Israel which leads to an unbalanced discussion for Eastern Mediterranean boundary layer. There are a number of boundary layer studies from other coastal regions in Eastern Mediterranean and their links to atmospheric pollution (e.g. Melas and Enger, 1993; Kallos et al., 1993; Svensson, 1996; Kostopoulos and Helmis, 2014; Tombrou et al., 2015).

7) The connection of the Sections 2.2, 2.3 and 2.4 with the discussion of the manuscript in later sections is fragile. The Sections 2.2, 2.3 and 2.4 could be merged into one broader in scope section for the role of the atmospheric boundary layer for atmospheric pollution over Eastern Mediterranean and make a stronger link with the core part of the review paper which is the atmospheric pollution concentration distribution during summer.

8) Section 2.5 in its current form provides basically information from a single study for a receptor site at Israel. It does not provide an overview over Eastern Mediterranean. I am not sure what is the added value of this Section in its current form.

9) Page 13, lines 17-19: The authors state that "in general , mineral dust does not affect the EM during summer". This is a rather strong statement. Consider that there a number of observational and modeling studies indicating a contribution of 25-30 % of dust aerosols on the total aerosol optical depth during summer over land and see in Eastern Mediterranean (Gerasopoulos et al., 2011; Georgoulias et al., 2016; Tsikerdekis et al., 2017; Marinou et al., 2017).

10) Page 15, lines 7-14: This is not exactly the finding of the study by Tyrlis and Lelieveld (2013). The various components observed over the Eastern Mediterranean that include the Etesians, subsidence, tropopause folds, stratospheric intrusions, and the summer ozone pool are dynamically interwoven manifestations of the influences induced by the South Asian monsoon and the midlatitudes. Tropopause folds and the

subsidence are the key components yielding high ozone concentrations in the middle and lower free troposphere over the region (see e.g. the recent publications on the topic by Tyrlis et al., 2014 and Akritidis et al., 2016).

11) Page 18, lines 19-20: " ... controlled by the strength of Azores High and the PT" . See my comment 3. There are a number of studies showing the differences between the acticylonic center over central and southeastern Europe causing the Etesians and the Azores High.

12) Section 3.2: The majority of the discussed articles refer to studies at the coast of Israel which leads to an unbalanced discussion for Eastern Mediterranean sulfate aerosols. Consider that there are some other earlier and recent studies for sulfate aerosols, SO2 and their transport over Eastern Mediterranean during summer (e.g. Mihalopoulos et al., 1997; Kouvarakis and Mihalopoulos; Zerefos et al., 2000; Kubilay et al., 2002; Karnieli et al., 2009; Georgoulias et al., 2009; 2016).

13) Section 3.3.: The discussion of the NOy species is fragmented, with lack of coherency and it does not provide a thorough overview over the Eastern Mediterranean regional baseline. In the begging there is some reference to baseline observational studies , then there is a sudden shift to a more extensive discussion of NOy and NOx species at urban sites at Israel and in the end there is a short discussion of satellite studies of tropospheric NO2 columnar densities.

14) Page 24, line 14: It is written that " NOy, the total reactive nitrogen (NO + NO2 + HNO3)..." . The NOy includes also PAN along with HNO4, N2O5, NO3 and other PAN homologues (PANs) and organic nitrates (Emmons et al., 1997).

15) Page 24, line 23: It is written that NO of 20 pptv were observed at Finokalia. This is rather low and not typical for Finokalia station. For example Kouvarakis et al., (2002) reports that NO concentrations ranged between the detection limit of 50 pptv (most of the time) and 100 pptv and NOx' between 0.1 and 4 ppbv. Also Gerasopoulos et al (2006) reports average day-time values of NO up to 80 pptv and respective NOz

values up to 1.6 ppbv. See also related articles for NOy measurements at Eastern Mediterranean from the MINOS campaign (Traub et al., 2003; Heland et al., 2003).

16) Section 3.4: The section focuses on CO sources and pathways but I think it is essential to give in the beginning also an overview of the CO baseline levels at Eastern Mediterranean based on observational studies. Furthermore the discussion is not balanced e.g. from page 27 (line 8) to page 29 (line 29) there is extending discussion on the results of a single article (Drori eta l., 2012)

17) Section 3.4: Methane is a long-lived species in contrast to all others species discussed earlier. I think the authors should make a distinction in the discussion of the short-lived pollutants versus the long-lived pollutants.

---

## Referee Comment (RC2) · Anonymous Referee #1 · 29 Jul 2017

Atmospheric pollution concentrations over the Eastern Mediterranean during summer – A review

Uri Dayan, Philippe Ricaud, Régina Zbinden and François Dulac

General Comments: This is a very carefully written paper reviewing extensively the literature on the concentration of pollutants in the Eastern Mediterranean. As the title suggests, this review is focusing explicitly during the summer months. I consider this as a limitation, whereas, in the way the paper is structured, it could be more general covering the entire year and not just the summer season. Indeed, a lot of the meteorological background provided applies to most of the year. However, the authors very carefully stress the summertime atmospheric patterns more, in their effort of course to justify the title. Saying the above, I do not have any objection to the authors' seasonal reference

to their topic. It is a targeted paper for the East Mediterranean and for summer. This is acceptable. My only suggestion to conclude this comment of mine, is to add one sentence in the beginning stressing why summer is more important to review than any other time of the year (this will justify the title also). There are several statements like "The Eastern Mediterranean (EM) is one of the regions in the world where elevated concentrations of primary and secondary gaseous air pollutants have been reported frequently, mainly in summer" and "observed in the summer over the EM are among the highest over the Northern Hemisphere" etc. Maybe it is useful to summarize all these reasons for choosing EM and summertime by saying something like "based on the above, …... the paper focuses explicitly on EM and summertime, because . . ..". The structure of the paper is quite logical and the flow in reading through it is quite smooth. The paper could benefit by adding an Appendix which will tabulate the abbreviations and notations used. Since this is a review paper, another suggestion is to have a table of contents at the start of the paper, if this is of course allowed by the Journal's style. Such a table should facilitate the reader in locating specific section of the paper. I am wondering whether the use of ppbv is compatible with the use pf ppb. In figure 14, for example, ppbv is used in the figure's legend, whereas ppb is used in the line above this figure. Please check throughout the text. If ppb implies by volume anyhow, please unify the notation of the units used. Complete all figure legends and Table headings with a period at the end, if this is the style of the journal. My impression is that the number of figures (27) is too large, but it is up to the Editor and the Publisher to accept this. Almost all of the figures are reproduced from published research. Finally, I was not able to cross-check the list of references and the citations in the text; I leave this to the authors and the Journal's editorial team.

Minor suggestions: Although the use of the English language, syntax, punctuation and grammar is excellent, there are a few minor issues that the authors may decide to take care of.

Page 14, line 5 missing quotes "hot pot" Page 15, line 17 "et al." Page 15, line 20:

"They. . ." Whom does it refer to, Myriokefalitakis or Daksalakis? Page 15 line 28: It is not appropriate, to my understanding to start a sentence with O3. You can rephrase it "Measurements of O3. . ...". Please check elsewhere too. Page 15 lines 28-30: . . ...performed over. . .. and from. . . . Syntax inconsistency? Page 16 lines 1-3. Do you imply that from these two sets of measurements (Finokalia and Thessaloniki) they have deduced a @maximum over . . ...the Aegean Sea". I cannot see how this can be deduced from just two measurements. The Aegean is a rather large sea to be so definite about finding a maximum. Page 16, line 11: put a comma after ppbv Page line 21-22: pollutants concentrations OR pollutant concentrations ? Please check for correct English. Page 29 line 8: "; and" please check Page 30 line 17, Starting a sentence with CH4 does not seems appropriate. This was mentioned again, but it is up to the editorial style to adopt it or not. Recommendation: The paper may be accepted for publication in Atmos. Chem. Phys., after a minor revision.

---

## Author Comment (AC1) · 26 Sep 2017

First, we would like to thank the Referee for his valuable and insightful comments which improved much this review study. Following are our detailed responses to each of the comment posted:

1) Section 2.1, page 4, line 15: The authors state that dry north Etesian winds are generated by the Persian Trough. It is actually generated by the east–west pressure gradient manifested by large scale circulation features, low pressures over eastern Mediterranean/Middle East as an extension of the PT and the high pressure over central and southeastern Europe.

We agree with the Referee, the text was corrected accordingly and referred to Tyrlis

and Lelieveld, 2013. .

2) Section 2.1, page 4, lines 16-21: The discussion for the eastern Mediterranean subsidence during summertime needs elaboration in connection to the discussion in page 5 (lines 16-19) based on the paper of Rodwell and Hoskins. The current consensus view recognizes the importance of the interaction with the mid-latitude westerlies of an equatorially trapped Rossby wave to its west induced by the South Asian monsoon heating as well as an enhancement of the descent due to diabatic radiative cooling under clear sky conditions (Rodwell and Hoskins 1996, 2001; Tyrlis et al. 2012).

We would like to thank the reviewer for this important comment. A full paragraph was added to the revised manuscript so as to explain better the summer thermodynamic and dynamic conditions and the important role of the South Asian monsoon on summer subsidence over the EM.

3) Section 2.1, page 4, line 25: The discussion for the eastward progression of the subtropical high needs clarification. Which subtropical high do the authors mean? During summer the Azores High moves westward toward Bermuda (when it is known as the Bermuda high). Furthermore a number of studies point out the differences between the acticylonic center over central and southeastern Europe causing the Etesians and the Azores permanent Anticyclone (Prezerakos, 1984; Tyrlis and Lelieveld, 2013; Anagnostopoulou et al., 2014). The acticylonic center over central and southeastern Europe causing the Etesians is related primarily with anticyclonic vorticity advection from Northwestern Africa and secondly with diabatic cooling under clear skies.

We agree with the Referee as regarded to the anticyclonic centers formed over the Balkans. A paragraph was added to the revised manuscript explaining why such centers cannot be considered as extensions of the Azores. A description of the dynamic conditions for their development is given.

4) Section 2.2, page 6, line 11: The authors state "Since this turbulent layer is mainly governed by synoptic-scale circulation patterns ..." What exactly do the authors mean?

[Figure]

Please clarify. Is this a general comment or a comment associated with the specific cited study of Dayan et al., 1988?

In order to better explain the role of the synoptic scale circulation on shaping the structure and depth of the atmospheric mixed layer, a short paragraph and references (Businger and Charnock, 1983; Holt and Raman, 1990; Sinclair et al., 2010) supporting such a statement was added to the text.

5) and 6): Sections 2.2: There is extensive description of the link between synoptic patterns and the structure of the mixing layer in Israel within this session. As a reader I am rather confused and I do not really see the scope of such extended description of this link for a specific region in the frame of an overview paper for the regional baseline atmospheric pollution concentrations over Eastern Mediterranean. The majority of the discussed articles refer to studies at the coast of Israel which leads to an unbalanced discussion for Eastern Mediterranean boundary layer. There are a number of boundary layer studies from other coastal regions in Eastern Mediterranean and their links to atmospheric pollution (e.g. Melas and Enger, 1993; Kallos et al., 1993; Svensson, 1996; Kostopoulos and Helmis, 2014; Tombrou et al., 2015).

We do agree with the Referee as regarded to the rather unbalanced description of the MLD in the EM by reviewing studies representing mostly the coast of Israel. Consequently, this whole section was rewritten while referring to other boundary layer studies from other coastal regions in the EM basin such as the Greek Peninsula Crete and Cyprus. (Kassomenos et al. 1995, Svensson 1996, Leventidu et al. 2013, Tombrou et al. 2015, Zbinden et al., 2016),

7) The connection of the Sections 2.2, 2.3 and 2.4 with the discussion of the manuscript in later sections is fragile. The Sections 2.2, 2.3 and 2.4 could be merged into one broader in scope section for the role of the atmospheric boundary layer for atmospheric pollution over Eastern Mediterranean and make a stronger link with the core part of the review paper which is the atmospheric pollution concentration distribution during

summer.

We adopted the suggestion made by the Referee and merged sections 2.2, 2.3 and 2.4 into one broader section entitled: "Atmospheric dispersion conditions over the EM". This new section describes the spatio-temporal distribution of the MLD and points on the differences observed in these characteristics among several sites within the EM basin. We believe that this modification indeed reinforces the link between both essential components of this review, the atmospheric processes and the issuing tropospheric pollutant concentrations.

8) Section 2.5: This section in its current form provides basically information from a single study for a receptor site at Israel. It does not provide an overview over Eastern Mediterranean. I am not sure what the added value of this Section is in its current form.

We do believe that the chemical composition of an air mass is inevitably related to its origins and pathways. Therefore, we think that such a section dealing with air mass origins is necessary. However, we fully agree that one sole study representing the flow climatology during summer over the EM is not sufficient. Accordingly, a paragraph was added to this section by describing the essential results that were obtained from other similar flow climatology studies conducted over Greece (Katsoulis 1999) and Turkey (Kubilay (1996) and compared them to the one performed over the central coast of Israel.

9) Page 13, lines 17-19: The authors state that "in general, mineral dust does not affect the EM during summer". This is a rather strong statement. Consider that there a number of observational and modeling studies indicating a contribution of 25-30 % of dust aerosols on the total aerosol optical depth during summer over land and see in Eastern Mediterranean (Gerasopoulos et al., 2011; Georgoulias et al., 2016; Tsikerdekis et al., 2017; Marinou et al., 2017).

We admit that our statement as regarded to the partial contribution of mineralogical dust to the EM during summer was expressed in a rather too strong and liberal manner.

Actually, besides the references given by the Referee on this issue, we ourselves were involved in such studies, e.g., Erel et al., 2007; Kalderon-Asael et al., 2009; Erel et al., 2013. The decision to limit ourselves only to the contribution of gaseous pollutants was derived from our awareness of the numerous studies published on the subject, in order not to create an overwhelming article. The relevant paragraph in the original manuscript has been moved to the Introduction and modified accordingly.

Ref: Erel, Y., B. Kalderon-Asael, U. Dayan, and A. Sandler (2007): European Pollution Imported by Cooler Air Masses to the Eastern Mediterranean during the Summer Environ. Sci. Technol. 41, 5198-5203.

Kalderon-Asael, B., Y.Erel, A. Sandler and U. Dayan (2009): Mineralogical and chemical characterization of suspended atmospheric particles over the East Mediterranean based on synoptic-scale circulation patterns. Atmospheric Environment, doi: 10.1016/j. atmosenv. 2009.03.057.

Erel, Y., O. Tirosh, N. Kessler, U. Dayan, S. Belkin, M. Stein, A. Sandler, and J.J. Schauer (2013): Atmospheric particulate matter (PM) in the Middle East: Toxicity, transboundary transport, and influence of synoptic conditions, in P. Censi et al. (eds.), Medical Geochemistry: Geological Materials and Health, DOI 10.1007/978-94-007-4372-4__3, © Springer Science C Business Media Dordrecht 2013.

10) Page 15, lines 7-14: This is not exactly the finding of the study by Tyrlis and Lelieveld (2013). The various components observed over the Eastern Mediterranean that include the Etesians, subsidence, tropopause folds, stratospheric intrusions, and the summer ozone pool are dynamically interwoven manifestations of the influences induced by the South Asian monsoon and the midlatitudes. Tropopause folds and the subsidence are the key components yielding high ozone concentrations in the middle and lower free troposphere over the region (see e.g. the recent publications on the topic by Tyrlis et al., 2014 and Akritidis et al., 2016).

On the key components yielding high ozone concentrations in the middle and lower free

troposphere over the region: We would like to thank the Referee for this constructive comment while driving our attention to further references enabling us to give a better description of the referenced interlaced dynamical processes yielding to the concentration measured and simulated. Accordingly, a full paragraph was added while referring to Tyrlis and Lelieveld, 2013 and Tyrlis et al., 2014).

11) Page 18, lines 19-20: "... controlled by the strength of Azores High and the PT". See my comment 3. There are a number of studies showing the differences between the acticylonic center over central and southeastern Europe causing the Etesians and the Azores High.

The text was corrected as regarded to the anticyclonic centers formed over the Balkans which generate the Etesians rather than the Azores High.

12) Section 3.2: The majority of the discussed articles refer to studies at the coast of Israel which leads to an unbalanced discussion for Eastern Mediterranean sulfate aerosols. Consider that there are some other earlier and recent studies for sulfate aerosols, SO2 and their transport over Eastern Mediterranean during summer (e.g. Mihalopoulos et al., 1997; Kouvarakis and Mihalopoulos; Zerefos et al., 2000; Kubilay et al., 2002; Karnieli et al., 2009; Georgoulias et al., 2009; 2016).

We agree that the survey on EM sulfate aerosols could be further broadened for a better-balanced presentation of this issue. Consequently, we revised and used the references offered by the Referee and others in order to enrich the discussion of this section.

13) 14) and 15) Section 3.3: The discussion of the NOy species is fragmented, with lack of coherency and it does not provide a thorough overview over the Eastern Mediterranean regional baseline. In the begging there is some reference to baseline observational studies, then there is a sudden shift to a more extensive discussion of NOy and NOx species at urban sites at Israel and in the end there is a short discussion of satellite studies of tropospheric NO2 columnar densities. Page 24, line 14: It is written that

" NOy, the total reactive nitrogen (NO + NO2 + HNO3)..." . The NOy includes also PAN along with HNO4, N2O5, NO3 and other PAN homologues (PANs) and organic nitrates (Emmons et al., 1997). Page 24, line 23: It is written that NO of 20 pptv were observed at Finokalia. This is rather low and not typical for Finokalia station. For example Kou-varakis et al., (2002) reports that NO concentrations ranged between the detection limit of 50 pptv (most of the time) and 100 pptv and NOx' between 0.1 and 4 ppbv. Also Gerasopoulos et al (2006) reports average day-time values of NO up to 80 pptv and respective NOz values up to 1.6 ppbv. See also related articles for NOy measurements at Eastern Mediterranean from the MINOS campaign (Traub et al., 2003; Heland et al., 2003).

We agree on the constructive comments made by the Referee. Consequently, the whole Section 3.3 was restructured. We referred to some more studies related to NOy measurements over Greece and Turkey along the references that were given by the Referee plus some others (Lammel and Cape, 1996; Amaroso et al., 2008; Im et al., 2008; and Ozden et al., 2008). Our referring to these additional studies, lead to a more balanced discussion reflecting better the whole EM region. Moreover, some parts of this section giving too many specific details of Israeli studies were significantly shortened. We do believe that this section in its reconstructed version gives a much more complete overview of the NOy baseline levels as observed over the EM during summer.

16) Section 3.4: The section focuses on CO sources and pathways but I think it is es-sential to give in the beginning also an overview of the CO baseline levels at Eastern Mediterranean based on observational studies. Furthermore the discussion is not bal-anced e.g. from page 27 (line 8) to page 29 (line 29) there is extending discussion on the results of a single article (Drori eta l., 2012).

As suggested by the Referee, we added a full paragraph to this section surveying sev-eral observational studies to get an insight of the CO baseline levels as reported from few EM countries (i.e., Greece (Riga-Karandinos and Saitanis, 2005), Lebanon (Saliba

et al. 2006) and the Gaza strip, Palestine (Elbayoumi et al. (2014). Furthermore, for a better balancing of this section, we reduced slightly the discussion on CO sources and pathways based mainly on the results of Drori et al. (2012) paper.

17) Section 3.5: Methane is a long-lived species in contrast to all others species discussed earlier. I think the authors should make a distinction in the discussion of the short-lived pollutants versus the long-lived pollutants.

Discussing Methane as a long-lived species: A full paragraph was added to this section in which the long lifetime nature of Methane in contrast to other trace gases was demonstrated by few studies dealing with its possible association to low frequency atmospheric circulation patterns (i.e., ENSO and NAO). Moreover, the differing lifetimes of the pollutants surveyed in this study and their issuing implications was mentioned in the Introduction section.

---

## Author Comment (AC2) · 26 Sep 2017

The authors of this study were pleased that the Referee was satisfied in the way this paper was written and considered carefully most of the comments he made.

General comments: 1) The suggestion to add one sentence in the beginning stressing why summer is more important to review than any other time of the year.

We fully agree with the Referee, accordingly, a paragraph enumerating the essential factors affecting the EM during summer was added at the beginning of the Introduction chapter.

2) The comment made by the Referee on adding an Appendix which will tabulate the abbreviations and notations used.

[Figure]

Since all terms are detailed in the main text, we believe that an Appendix tabulating the abbreviations used in the text is redundant and was therefore not added to the manuscript as suggested by the Referee.

3) The suggestion made by the Referee on adding a Table of Content at the beginning of the paper, if allowed by the Journal's style.

We checked this issue with an Editorial representative of Copernicus Publications and received a negative reply while mentioning that a table of contents should not be included according to their house standards.

4) The comment made on unifying the notation of the PPB and PPBV units used.

We agree with the Referee. PPBV are the volumetric mixing ratios. The notation was unified and corrected throughout the text as well as in Figs. 14, 19, 20, and 21.

5) Comment made by the Referee on the necessity to complete all figure legends and Table headings with a period at the end.

Done.

Minor suggestions:

1) Page 14, line 5 missing quotes "hot pot"; Page 15, line 17 "et al.".

Done.

2) Page 15, line 20: Appropriate reference.

The Referee is right, we apologize for the confusion. The sentence was corrected and the reference of Daskalakis et al., 2015 was omitted.

3) Page 15, line 28: inappropriateness to start a sentence with $O_3$ .

We agree with the Referee and rephrased this sentence with "Measurements of $O_3$"

4) Page 15 lines 28-30 and Page 16 lines 1-3: Comment made by the Referee on

implying on the summer O3 maximum observed over the Aegian Sea and Finokalia based on just two sets of measurements (Finokalia and Thessaloniki).

We fully agree that this paragraph was expressed in a rather confusing manner. Inferring on the summer maximum O3 over the Aegean Sea and Finokalia, Crete was based on several measurements and for rather long periods: 1) An O3 analyzer installed in a vessel traveling routinely from Heraklion, Crete to Thessaloniki, Greece, from Aug. to Nov. 2000, 2) From the regional monitoring station of Finokalia, Crete from Sept.1997 to Sept. 1999 (Kouvarakis et al. 2000) and 3) From a site nearby Thessaloniki (40° 32' N, 23° 50' E) from Mar. 2000 to Jan. 2001. The text was corrected accordingly.

Ref: Kouvarakis, G., K. Tsigaridis, M. Kanakidou, and N. Mihalopoulos, Temporal variations of surface regional background ozone over Crete Island in southeast Mediterranean, J. Geophys. Res., 105, 4399 – 4407, 2000.

5) Page 16, line 11: put a comma after ppbv.

Done.

6) Page line 21-22: pollutants concentrations OR pollutant concentrations?

The right term is "air pollutant concentrations". This term was applied throughout the text.

7) Page 29 line 8: omit coma.

Done.

8) Page 30 line 17: Starting a sentence with CH4 does not seem appropriate.

We tend to agree with the Referee, and started this paragraph as follows: "Methane (CH4) is the most abundant hydrocarbon in the atmosphere. . .."
* * *